# Mutually Regressive Point Processes

**Ifigeneia Apostolopoulou**
Machine Learning Department
Carnegie Mellon University
iapostol@andrew.cmu.edu

**Scott Linderman**
Department of Statistics
Stanford University
scott.linderman@stanford.edu

**Kyle Miller**
AutonLab
Carnegie Mellon University
mille856@andrew.cmu.edu

**Artur Dubrawski**
AutonLab
Carnegie Mellon University
awd@cs.cmu.edu

## Abstract

Many real-world data represent sequences of interdependent events unfolding over time. They can be modeled naturally as realizations of a point process. Despite many potential applications, existing point process models are limited in their ability to capture complex patterns of interaction. Hawkes processes admit many efficient inference algorithms, but are limited to mutually excitatory effects. Nonlinear Hawkes processes allow for more complex influence patterns, but for their estimation it is typically necessary to resort to discrete-time approximations that may yield poor generative models. In this paper, we introduce the first general class of Bayesian point process models extended with a nonlinear component that allows both excitatory and inhibitory relationships in continuous time. We derive a fully Bayesian inference algorithm for these processes using Pólya-Gamma augmentation and Poisson thinning. We evaluate the proposed model on single and multi-neuronal spike train recordings. Results demonstrate that the proposed model, unlike existing point process models, can generate biologically-plausible spike trains, while still achieving competitive predictive likelihoods.

## 1 Introduction

Many natural phenomena and practical applications involve asynchronous and irregular events such as social media dynamics, neuronal activity, or high frequency financial markets [1, 2, 3, 4, 5, 6, 7, 8]. Modeling correlations between events of various types may reveal informative patterns, help predict next occurrences, or guide interventions to trigger or prevent future events. *Point Processes* [9] are models for the distribution of sequences of events.

*Cox processes* or *doubly stochastic processes* [10] are generalizations of *Poisson Processes* [11], where the intensity function is a stochastic process itself. Although there are efficient inference algorithms for some of their variants [12, 13], Cox processes do not capture explicitly temporal correlations between historical and future events. On the other hand, the *Hawkes Process (HP)* [14, 15] and its variants [16, 17, 18] constitute a class of point process models where past events linearly combine to increase the probability of future events. However, purely excitatory effects are incapable of characterizing physiological patterns such as neuronal activity where inhibitory effects are present and crucial for self-regulation [19, 20, 21]. The work in [22] can support temporal effects beyond mutual excitation that HP misses. However, capturing model uncertainty is critical in many applications [23, 24, 25, 26], especially when the size of the available data is limited compared to the model complexity. Rich literature exists on HP-based learning tasks [27, 28, 29, 30, 31, 32].

A nonlinear generalization of the HP allows for both excitatory and inhibitory interactions, but evaluating the probability density of these models requires computing integrated intensity, which is

generally intractable. Instead, we are forced to use discrete time approximations, which reduce to a *Poisson Generalized Linear Model (Poisson-GLM)* [33, 3], making learning of these models from data very efficient. However, the estimated regression coefficients may vary widely depending on the boundaries chosen for aggregation [34]. Empirical evidence suggests that while suitable for one-step predictions, such models may suffer stochastic instability and yield non-physical predictions [35].

There is currently limited statistical theory for point process models that support complex temporal interactions in a continuous-time regime. To this end, we develop the first class of Bayesian point process models—*Mutually Regressive Point Processes (MR-PP)*—that allow for nonlinear temporal interactions while still admitting an efficient, fully-Bayesian inference algorithm in continuous time.

## 2 Proposed Model

### 2.1 Problem statement

We are interested in learning distributions over event sequences (point processes). These distributions are mutually regressive in the sense that past event occurrences can influence the future realization of the process in an arbitrary manner. A Point Process $\mathcal{PP}(\lambda(t))$ is characterized by an intensity function $\lambda(t)$, so that in an infinitesimally wide interval $[t, t + dt]$, the probability of the arrival of a new event is $\lambda(t)dt$ [36].

### 2.2 Classical Hawkes Process

A Hawkes process (HP) [14, 15] of $N$ event types $\mathcal{HP}_N(\lambda_n^*(t))$ is characterized by the intensity functions $\lambda_n^*(t)$ for the events of type $n$ defined as:

$$\lambda_n^*(t) = \lambda_n^* + \sum_{m=1}^{N} \sum_{i=1}^{K_m} \lambda_{m,n}(t, t_i^m)\mathcal{I}(t_i^m < t), \tag{1}$$

$$\lambda_{m,n}(t, t_i^m) = \alpha_{m,n}\, e^{-\delta_{m,n}(t-t_i^m)}, \tag{2}$$

where $\lambda_n^* \geq 0$, $\alpha_{m,n} \geq 0$, and $\delta_{m,n} > 0$. $t_i^m$ is the arrival time of the $i$-th event of type $m$ and $K_m$ is the number of events of type $m$. $\mathcal{I}$ is the indicator function. By the superposition theorem for Poisson processes, the additive terms in Equation (1) can be viewed as the superposition of independent *non-homogeneous* Poisson processes (with intensity function that varies in time) characterized by the intensity functions $\lambda_{m,n}(t, t_i^m)$, triggered by the event $i$-th of type $m$ that occurred before time $t$, and an exogenous, *homogeneous* Poisson process characterized by the constant intensity function $\lambda_n^*$. The HP is a *mutually exciting* point process in the sense that past events can only raise the probability of arrival of future events of the same or different type. Since $\lambda_n^*(t)$ depends on past occurrences, it is a stochastic process itself.

### 2.3 Mutually Regressive Point Process: a generalization of the Hawkes Process

The intensity function $\lambda_n(t)$, for events of type $n$ occurring at times $\dot{t}_i^n$, of a Mutually Regressive Point Process (MR-PP) is a HP intensity augmented with a probability term. It is defined as follows:

$$\lambda_n(t) = \lambda_n^*(t)p_n(t), \tag{3}$$

$$\lambda_n^*(t) = \lambda_n^* + \sum_{m=1}^{N} \sum_{i=1}^{K_m} \lambda_{m,n}(t, \dot{t}_i^m)\mathcal{I}(\dot{t}_i^m < t), \tag{4}$$

$$p_n(t) = \sigma(\boldsymbol{w}_n^T \boldsymbol{h}(t)), \tag{5}$$

$$h_m(t) = c \sum_{i=1}^{K_m} h(t, \dot{t}_i^m)\mathcal{I}(\dot{t}_m^i < t), \tag{6}$$

$$h(t, \dot{t}_i^m) = e^{-\gamma(t-\dot{t}_i^m)}, \tag{7}$$

where $\lambda_n^* \geq 0$, $c > 0$, $\gamma > 0$, $\boldsymbol{w}_n = [b_n, w_{1,n}, w_{2,n}, \ldots, w_{N,n}]^T$, $\boldsymbol{h}(t) = [1, h_1(t), h_2(t), \ldots, h_N(t)]^T$ and $\lambda_{m,n}(t, \dot{t}_i^m)$ defined in Equation (2). $\sigma(x) = (1 + e^{-x})^{-1}$ is the sigmoid function. The weight $w_{m,n}$ models the influence of type $m$ on type $n$ and $h_m(t)$ is the

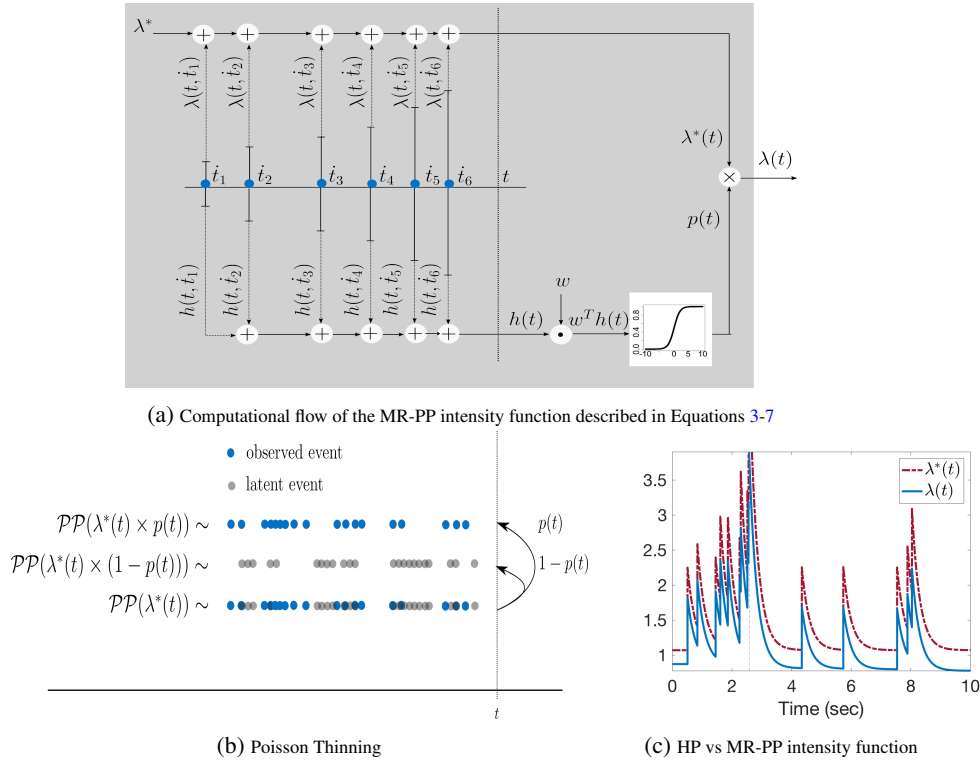

(a) Computational flow of the MR-PP intensity function described in Equations 3-7

(b) Poisson Thinning

(c) HP vs MR-PP intensity function

Figure 1: *Explanation of the MR-PP.* The computation of the intensity function of a MR-PP at time $t$ as a function of the past events is explained in Figure 1a. Figure 1b shows the simulation of a MR-PP which can be viewed as classification of events generated by a HP as either latent or observed. The point processes of the observed and thinned events are characterized by the $\lambda^*(t)$ intensity multiplied by the probability term $p(t)$ and $1 - p(t)$ respectively. The upper-bounding, mutually exciting, intensity and the thinned intensity $\lambda(t) = \lambda^*(t) \times p(t)$ which generates the observed events is shown in 1c.

aggregated temporal influence of type $m$ up to time $t$. The computational procedure is illustrated in Figure 1a. The effect of the probability term on the upper-bounding intensity $\lambda_n^*(t)$ is demonstrated in Figure 1c. We can simulate from this model via Poisson thinning [37, 12]. First, we sample N sets of events $t_1^n, t_2^n, \ldots,$ for $n = 1, 2, \ldots, N$, from a $\mathcal{HP}_N(\lambda_n^*(t))$. Afterwards, we chronologically proceed through the simulated events and accept them with probability $\lambda_n(t)/\lambda_n^*(t) = p_n(t)$, the relative intensity at that point in time (Figure 1b). In case an event at $t_i^n$ is rejected, its offsprings (events generated by $\lambda_{n,m}(t, t_i^n)$) are pruned so that the $\lambda_n^*(t)$ defined in Equation (4) depends only on the realized events whose arrival times are notated as $\dot{t}_i^m$. Importantly, the relative intensity $p_n(t)$ and the intensity $\lambda_n^*(t)$ only depend on the preceding events that were *accepted*; rejected events have no influence on the future intensity. Note that a negative weight $w_{m,n}$ means that events of type $m$ inhibit future events of type $n$ since $h_m(t)$ decreases $p_n(t)$. The correctness of this procedure is provided in the Supplementary Material.

Although $\lambda_n^*(t)$ could be replaced by a homogeneous Poisson intensity $\lambda_n^*$ so that any excitatory relationships are captured by a positive weight $w_{m,n}$, the upper bound $\lambda_n^*$ should be given a very large value in cases where the underlying process exhibits sparse event bursts. This fact, in turn, could yield a large number of latent events and hence render the learning of the model computationally intractable (see Section 3.1 for details). Moreover, MR-PP is not hardwired to exponential kernels. Alternative kernel functions, such as the Power-Law or the Rayleigh function could be used in Equations (2) and (7).

## 2.4 Hierarchical MR-PP for relational constraints

A dependence between the parameters of the intensity $\lambda_n^*(t)$ and the thinning procedure $p_n(t)$ can be imposed so that an interaction between types $m$ and $n$ is either inhibitory or excitatory (but not both) in a probabilistic manner. To this end, we define a *Sparse Normal-Gamma* prior for the weights,

which fosters an inverse relationship between the excitatory effect $\alpha_{m,n}$ and the repulsive effect $w_{m,n}$ of type $m$ on type $n$. It is motivated by the framework of *Sparse Bayesian Learning* [38, 39], in the sense that it associates an individual precision $\tau_{m,n}$ and mean $\mu_{m,n}$ with each weight $w_{m,n}$. $\mu_{m,n}$ and $\tau_{m,n}$ follow a Normal-Gamma distribution that depends on $\alpha_{m,n}$. It is defined as follows:

$$\tau_{m,n} \sim \text{Gamma}(\nu_\tau \phi_\tau(\alpha_{m,n}) + \alpha_\tau, \beta_\tau), \tag{8}$$

$$\mu_{m,n} \sim \mathcal{N}(-(\nu_\mu \phi_\mu(\alpha_{m,n}) + \alpha_\mu)^{-1}, (\lambda_\mu \tau_{m,n})^{-1}), \tag{9}$$

$$\boldsymbol{w}_n \sim \mathcal{N}(\boldsymbol{\mu}_n, \boldsymbol{\Sigma}_n), \tag{10}$$

where $\nu_\tau > 0$, $\alpha_\tau > 0$, $\beta_\tau > 0$, $\nu_\mu > 0$, $\alpha_\mu \geq 0$, $\lambda_\mu > 0$, $\boldsymbol{\mu}_n = [\mu_0, \mu_{1,n}, \mu_{2,n}, \ldots, \mu_{N,n}]^T$, $\boldsymbol{\tau}_n = [1/\sigma_0{}^2, \tau_{1,n}, \tau_{2,n}, \ldots, \tau_{N,n}]^T$, and $\boldsymbol{\Sigma}_n = diag(\boldsymbol{\tau}_n)^{-1}$. $\phi_\tau(x)$ and $\phi_\mu(x)$ are monotonically increasing positive activation functions.

A suggested activation function for $\tau_{m,n}$ and $\mu_{m,n}$ is a shifted and scaled sigmoid function which has a soft-thresholding effect:

$$\phi(\alpha_{m,n}) = \frac{1}{1 + e^{-\delta_0(\alpha_{m,n} - \alpha_0)}}. \tag{11}$$

$\alpha_0 > 0$ can be viewed as the excitation threshold (so that values of $\alpha_{m,n}$ above $\alpha_0$ indicate an excitatory relationship) and $\delta_0 > 0$ regulates the smoothness of the thresholding.

Note that when $\alpha_{m,n}$ is large (there is excitatory relationship from type $m$ on type $n$), the precision $\tau_{m,n}$ becomes large (approximately drawn from $\text{Gamma}(\nu_\tau, \beta_\tau)$) assuming $\nu_\tau >> \alpha_\tau$ and $\nu_\tau >> \beta_\tau$. Therefore, the variance $\tau_{m,n}^{-1}$ has a value close to zero with high probability. A similar scenario holds for $\mu_{m,n}$ if $\nu_\mu >> \alpha_\mu$. A small mean and variance for $w_{m,n}$ implies that any additional (possibly inhibitory) effect of type $m$ on type $n$ is suppressed. A numerical example is given in Figure 2a. On the other hand, when $\alpha_{m,n}$ is small, the precision of $\tau_{m,n}$ will take a small value approximately drawn from $\text{Gamma}(\alpha_\tau, \beta_\tau)$ (assuming that $\nu_\tau \phi_\tau(\alpha_{m,n}) << \alpha_\tau$ and $\alpha_\tau < \beta_\tau$). Similarly, $\mu_{m,n}$ can take large negative values coming approximately from a Normal distribution with mean $-\alpha_\mu^{-1}$. As a consequence, inhibitory effects from type $m$ on type $n$ are enabled. A numerical example is given in Figure 2b.

Due to the inverse relationship between the inhibitory coefficients $w_{m,n}$ and the endogenous intensity rates $\alpha_{m,n}$, relational constraints on pairs of types are established. Intuitively, the constants $\nu_\tau$, $\nu_\mu$ control the strength of these constraints, so that $w_{m,n}$ is close to zero for a large $\alpha_{m,n}$ with an adjustable probability. A traditional Hawkes process can be obtained by setting $\nu_\tau$, $\nu_\mu$ $\lambda_\mu$, $\alpha_\tau$, $\alpha_\mu$, $\mu_0$ and $\tau_0$ to a very large value.

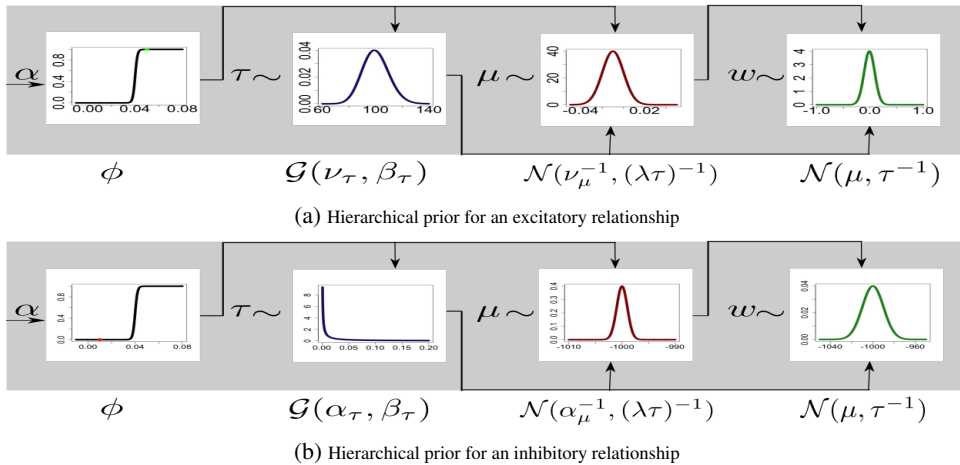

(a) Hierarchical prior for an excitatory relationship

(b) Hierarchical prior for an inhibitory relationship

Figure 2: *Illustration of the behavior of the hierarchical prior for enforcing relational constraints.* In 2a the excitatory coefficient $\alpha$ is above the threshold value (0.05) indicating an excitatory relationship. The prior drives the weights to a value close to zero. In 2b the coefficient is below the threshold indicating an inhibitory relationship. The prior steers the weights to a large negative value. The parameters of the hierarchical prior were set as follows: $\nu_\tau = 100, \alpha_\tau = 0.01, \beta_\tau = 1, \alpha_\mu = 0.001, \nu_\mu = 100, \lambda_\mu = 100$.

# 3 Bayesian Inference via Augmentation and Poisson Thinning

Here, we provide the description of the main components of the Bayesian inference for learning a MR-PP. It is also summarized in Algorithm 1. Full technical details are relegated to the Supplementary Material.

## 3.1 Generating latent events for tractability

The likelihood of the sequence $\mathcal{T} \triangleq \{t_i\}_{i=1}^{K}$ of $K$ events generated by a point process $\mathcal{PP}(\lambda(t))$ with intensity function $\lambda(t)$ in the time window $[0, T]$ is [36]:

$$p(\mathcal{T} \mid \lambda(t)) = \exp\left\{-\int_0^T \lambda(t)\, \mathrm{d}t\right\} \prod_{i=1}^{K} \lambda(t_i). \tag{12}$$

However, due to the sigmoid term in the intensity function described in Equations (3), (5), the integral and therefore sampling from posteriors which contain it, is intractable [12, 13]. This difficulty can be overcome by data augmentation [12], in which we jointly consider observed and thinned events akin to the Poisson thinning based sampling procedure mentioned in Section 2.3.

Let $\tilde{\mathcal{T}}_n \triangleq \{\tilde{t}_i^n\}_{i=1}^{M_n}$ be the sequence of $M_n$ latent (thinned) events of type $n$ and $\dot{\mathcal{T}}_n \triangleq \{\dot{t}_i^n\}_{i=1}^{K_n}$ be the $K_n$ observed events generated by thinning the process $\mathcal{PP}(\lambda_n^*(t))$ defined in Equation (4) by the probability $1 - p_n(t)$ and $p_n(t)$ respectively, where $p_n(t)$ is defined in Equation (5). Define the merged event sequence to be the ordered set:

$$\mathcal{T}_n \triangleq \dot{\mathcal{T}}_n \cup \tilde{\mathcal{T}}_n = \{t_i^n\}_{i=1}^{K_n+M_n}. \tag{13}$$

The joint likelihood of the arrival times along with the outcome of the Poisson thinning is then:

$$p(\mathcal{T}_n, \{s_i^n\}_{i=1}^{K_n+M_n} \mid \lambda_n^*(t), p_n(t)) =$$
$$\exp\left\{-\int_0^T \lambda_n^*(t)\, \mathrm{d}t\right\} \times \prod_{i=1}^{K_n+M_n} \lambda_n^*(t_i^n) \times \prod_{i=1}^{M_n+K_n} p_n(t_i^n)^{s_i^n} (1 - p_n(t_i^n))^{1-s_i^n}, \tag{14}$$

where $s_i^n \triangleq \mathcal{I}(t_i^n \in \dot{\mathcal{T}}_n) \in \{0, 1\}$ is the label indicating whether the event at $t_i^n$ is realized (belongs to $\dot{\mathcal{T}}_n$) or thinned (belongs to $\tilde{\mathcal{T}}_n$). Given Equation (14), the integral in the exponential term does not involve the sigmoidal term induced by $p_n(t)$. Therefore, efficient inference for the model parameters is feasible and it is reduced to the joint task of learning a Bayesian HP [40] and solving a Bayesian binary logistic regression (see Section 3.2).

## 3.2 Learning the nonlinear temporal interactions via Pólya-Gamma augmentation

The inference of the weights $w_{m,n}$ of the thinning procedure dictated by $p_n(t)$ amounts to solving a binary logistic regression problem for classifying the events as realized or thinned. From Equations (5), (10) and (14), and by keeping only the terms of the likelihood which contain $\boldsymbol{w}_n$, the posterior is obtained:

$$p(\boldsymbol{w}_n \mid \dots) \propto \mathcal{N}(\boldsymbol{w}_n; \boldsymbol{\mu}_n, \boldsymbol{\Sigma}_n) \times \prod_{i=1}^{K_n+M_n} \frac{e^{(\boldsymbol{w}_n^T \boldsymbol{h}(t_i^n)) \times s_i^n}}{e^{\boldsymbol{w}_n^T \boldsymbol{h}(t_i^n)} + 1}, \tag{15}$$

where we have used the property $1 - \sigma(x) = \sigma(-x)$. Sampling from this posterior can be done effeciently via Pólya-Gamma augmentation as in [e.g. 41, 42, 43, 13]. According to Theorem 1 in [41], the likelihood contribution of the thinning acceptance/ rejection of an event at time $t_i^n$ can be rewritten as:

$$\frac{e^{(\boldsymbol{w}_n^T \boldsymbol{h}(t_i^n)) \times s_i^n}}{e^{\boldsymbol{w}_n^T \boldsymbol{h}(t_i^n)} + 1} \propto \exp(\nu_i^n \boldsymbol{w}_n^T \boldsymbol{h}(t_i^n)) \times \int_0^\infty \exp\left\{-\frac{1}{2}\omega_i^n (\boldsymbol{w}_n^T \boldsymbol{h}(t_i^n))^2\right\} \mathcal{PG}_m(\omega_i^n; 1, 0)\, d\omega_i^n, \tag{16}$$

where $\nu_i^n = s_i^n - 1/2$, and $\mathcal{PG}_m(\omega_i^n; 1, 0)$ is the density of a Pólya-Gamma distribution with parameters $(1, 0)$. Combined with a prior on $\boldsymbol{w}_n$, the integrand in Equation (16) defines a joint density on $(s_i^n, \omega_i^n, \boldsymbol{w}_n)$, where $\omega_i^n$ is a latent Pólya-Gamma random variable. The posterior conditioned on

---

**Algorithm 1** Bayesian Inference for Mutually Regressive Point Processes

---

1. **Input**: Sequences of observed events $\{\dot{\mathcal{T}}_n\}_{n=1}^N$.

2. **Output**: Samples from $p\big(c, \gamma, \{\lambda_n^*, \boldsymbol{w}_n, \{\alpha_{m,n}, \delta_{m,n}\}_{m=1}^N\}_{n=1}^N \mid \{\dot{\mathcal{T}}_n\}_{n=1}^N\big)$.

3. Initialize randomly the model parameters from the priors.

4. **Repeat**

    (a) Sample the thinned events of type $n$ via Poisson thinning, for $n = 1, 2, \ldots, N$:

        i. from the exogenous intensity: $\tilde{\mathcal{T}}_n \sim \mathcal{PP}(\lambda_n^* (1 - p_n(t)))$, and

        ii. from the Poisson processes triggered by the observed events:

$$\big\{\{\tilde{\mathcal{T}}_n \sim \mathcal{PP}\big(\lambda_{m,n}(t - \dot{t}_i^m)(1 - p_n(t))\big)\}_{i=1}^{K_m}\big\}_{m=1}^N.$$

    (b) Sample the latent Pólya-Gamma variables of the observed and latent events:

$$\big\{\{\omega_i^n \sim \mathcal{PG}_m(1, \boldsymbol{w}_n^T \boldsymbol{h}(t_i^n))\}_{i=1}^{K_n+M_n}\big\}_{n=1}^N \text{ (Eq 21)}.$$

    (c) Jointly sample the weight prior parameters and the excitation coefficients $\{\alpha_{m,n}, \mu_{m,n}, \tau_{m,n}\}_{m,n=1}^N$ via collapsed Metropolis-Hastings.

    (d) Sample the weights for $n = 1, \ldots, N$: $\boldsymbol{w}_n \sim \mathcal{N}(\tilde{\boldsymbol{\Sigma}}_n, \tilde{\boldsymbol{\mu}}_n)$ (Eq 17, 18, 19 & 20).

    (e) Sample the rest of the parameters $c, \gamma, \{\lambda_n^*, \{\delta_{m,n}\}_{m=1}^N\}_{n=1}^N$.

---

the latent $\omega_i^n$ random variables becomes:

$$p(\boldsymbol{w}_n \mid \ldots) = \mathcal{N}(\boldsymbol{w}_n; \tilde{\boldsymbol{\Sigma}}_n, \tilde{\boldsymbol{\mu}}_n), \tag{17}$$

$$\text{where} \quad \tilde{\boldsymbol{\Sigma}}_n = \big(\boldsymbol{\Sigma}_n^{-1} + \boldsymbol{H}_n^T \boldsymbol{\Omega}_n \boldsymbol{H}_n\big)^{-1}, \quad \tilde{\boldsymbol{\mu}}_n = \tilde{\boldsymbol{\Sigma}}_n\big(\boldsymbol{\Sigma}_n^{-1}\boldsymbol{\mu}_n + \boldsymbol{H}_n^T \boldsymbol{\Omega}_n \boldsymbol{z}_n\big), \tag{18}$$

$$\text{and} \quad \boldsymbol{H}_n = [\boldsymbol{h}(t_1^n), \ldots, \boldsymbol{h}(t_{K_n+M_n}^n)]^T, \quad \boldsymbol{\Omega}_n = \mathrm{diag}(\omega_1^n, \ldots, \omega_{K_n+M_n}^n), \tag{19}$$

$$\boldsymbol{z}_n = \left[\frac{\nu_1^n}{\omega_1^n}, \ldots, \frac{\nu_{K_n+M_n}^n}{\omega_{K_n+M_n}^n}\right]^T. \tag{20}$$

From Theorem 1 in [41], for $\alpha = 1$ and $\beta = 1$, the posterior for sampling $\omega_i^n$ is

$$p(\omega_i^n \mid \ldots) = p(\omega_i^n \mid \{\dot{\mathcal{T}}_{n'}\}_{n'=1}^N, \boldsymbol{w}_n, c, \gamma) = \mathcal{PG}_m(\omega_i^n; 1, \boldsymbol{w}_n^T \boldsymbol{h}(t_i^n)). \tag{21}$$

### 3.3 Gibbs updates for the weights' prior mean and precision, and the intensity parameters

Since only one sample $w_{m,n}$ for sampling the mean $\mu_{m,n}$ and the precision $\tau_{m,n}$ is available, directly sampling from the posterior $p(\mu_{m,n}, \tau_{m,n}|w_{m,n}, \alpha_{m,n})$ would lead to poor mixing. This is also the case for sampling $\alpha_{m,n}$ from $p(\alpha_{m,n}|\mu_{m,n}, \tau_{m,n}, \ldots)$. Therefore, a joint collapsed Metropolis-Hastings update is used for sampling the excitation coefficient $\alpha_{m,n}$ and the weights' prior parameters $\mu_{m,n}$ and $\tau_{m,n}$, where the weight $w_{m,n}$ is collapsed. This is a similar in spirit to the technique in [38], where a collapsed likelihood is maximized. The collapsed Metropolis-Hastings ratio is derived in the Supplementary Material.

Given the observed and thinned events, conjugate updates are possible for the exogenous intensities $\lambda_n^*$ assuming a Gamma prior, a cluster-based Hawkes process representation [15] and by incorporating latent parent variables for the observed events [44, 45]. This is also the case for $\alpha_{m,n}$ in case of a flat MR-PP (defined in Section 2.3). The rest of the parameters are updated via adaptive Metropolis similar to [40]. The suggested proposal distributions and the Metropolis-Hastings ratios are given in the Supplementary Material.

## 4 Experimental Results [1]

### 4.1 Synthetic validation

We test our model and inference algorithm on synthetic data to ensure that we can recover the underlying interactions. We generated a MR-PP of two event types with parameters drawn from their priors (see Supplementary Material for the details) and we simulated it in the interval $[0, 20000]$.

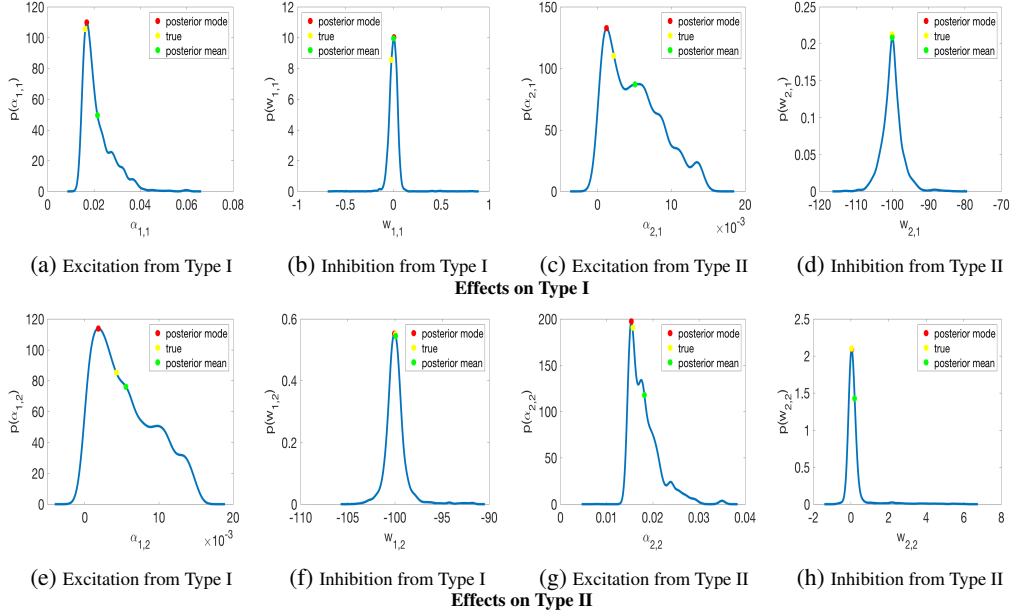

(a) Excitation from Type I    (b) Inhibition from Type I    (c) Excitation from Type II    (d) Inhibition from Type II

**Effects on Type I**

(e) Excitation from Type I    (f) Inhibition from Type I    (g) Excitation from Type II    (h) Inhibition from Type II

**Effects on Type II**

Figure 3: *Posterior distributions of the parameters of the synthetic MR-PP.* There is self-excitation and mutual inhibition for both types. The self-excitation is indicated by the large endogenous intensity rates $a_{1,1}$(3a) and $a_{2,2}$(3g) and the small weights $w_{1,1}$ (3b) and $w_{2,2}$ (3h). The mutual inhibition is indicated by the small $a_{2,1}$(3c) and $a_{1,2}$(3e) and the large negative $w_{2,1}$(3d) and $w_{1,2}$(3f). The correct interactions were discovered.

The derived synthetic dataset consists of 269 observed events that were used for the training. Type I excites events of Type I and inhibits events of Type II. Similarly, Type II inhibits events of Type I and excites events of Type II.

In Figures 3a-3d, we plot the posterior distribution, as well as the posterior mode and mean point estimates for the parameters $\alpha_{1,1}$, $w_{1,1}$ (temporal effect from Type I on Type I), and $\alpha_{2,1}$, $w_{2,1}$ (temporal effect from Type II on Type I). Both the real and the point estimates for the excitatory effect $\alpha_{1,1}$ (Figure 3a) from Type I are large (above the $\alpha_0 = 0.015$ threshold) compared to the suppressed, close to zero, weight $w_{1,1}$ (Figure 3b) indicating an excitatory relationship relationship. On the other hand, as shown in Figure 3d, the weight $w_{2,1}$ has a large negative value in contrast to $\alpha_{2,1}$ (Figure 3c), which has a close to zero value, indicating a repulsive relationship. A symmetric case of self-excitation (Figures 3g, 3h) and inhibition from the other type (Figures 3e, 3f) holds for Type II. Figure 4 shows the predictive log-likelihood for 1,000 held-

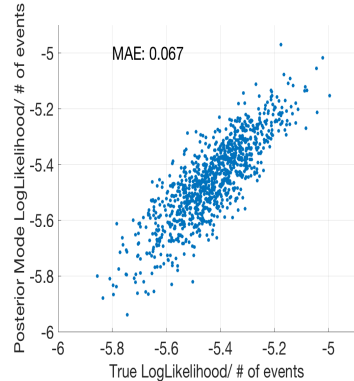

Figure 4: *Testing of the learned MR-PP on the synthetic data.* The scatterplot compares the log-likelihood for 1000 held-out event sequences of the true vs the learned MR-PP.

out event sequences with the real model parameters in contrast to that achieved by the posterior mode estimates, and the mean absolute error (MAE). The autocorrelation plots, the values of the hyperparameters and the learning parameters are provided in the Supplementary Material.

## 4.2 Experimental results on the stability of single neuron spiking dynamics

In this section, we study the quality of the MR-PP as a generative model. Although Point Process - Generalized Linear Models (PP-GLMs) have been extensively applied to a wide variety of spiking neuron data [3, 33, 46], they may yield non-physiological spiking patterns when simulated and used as generative models because of explosive firing rates although they pass goodness-of-fit tests [35, 47]. This could be potentially attributed to the fact that the excitatory properties are captured by non-linear terms in the model [48]. On the other hand, MR-PP inherently circumvents this by decoupling the linear excitatory portion from the non-linear but unit-bounded, inhibitory portion of the model. We repeat the analysis on two datasets (Figure 2.b and Figure 2.c in [35]) for which PP-GLMs have failed in generating stable spiking dynamics.

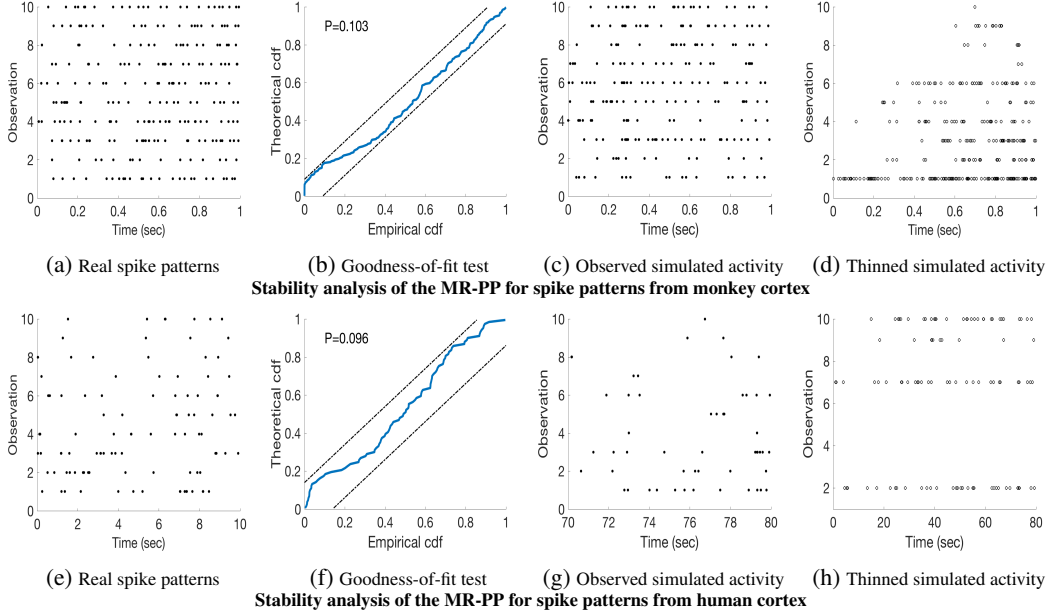

(a) Real spike patterns   (b) Goodness-of-fit test   (c) Observed simulated activity   (d) Thinned simulated activity

**Stability analysis of the MR-PP for spike patterns from monkey cortex**

(e) Real spike patterns   (f) Goodness-of-fit test   (g) Observed simulated activity   (h) Thinned simulated activity

**Stability analysis of the MR-PP for spike patterns from human cortex**

Figure 5: *Stability analysis of the MR-PP for cortex spike patterns.* In Figures 5a- 5d, we repeat the analysis of Figure 2.c for monkey cortex spike trains, and in Figures 5e- 5h, we repeat the analysis of Figure 2.b for human cortex spike train in [35]. In contrast to the PP-GLM, MR-PP both passes the goodness-of-fit test (5b),(5f) and generates stable spike trains (5c),(5g) similar to those used for the learning (5a),(5e).

Figure 5a illustrates ten 1-second observations of single-neuron activity from monkey area PMv cortical recordings used in [35]. We fit the MR-PP and we applied the time-rescaling theorem [49, 50] on the learned intensities and the real spike sequences. According to it, the realization of the general temporal point process can be transformed to one of a homogeneous Poisson process with unit intensity rate. Therefore, the well-studied Kolmogorov-Smirnov (KS) test can be capitalized for the comparison of the rescaled interspike arrivals to the exponential. Figure 5b shows the KS plot as in [49] for comparison of the empirical with the exponential distribution. The MR-PP passes the goodness-of-fit test ($p - value > 0.05$). Finally, we simulated the learned MR-PP for 1 second. Figure 5c shows the observed events of the process. The simulated activity of the learned MR-PP shown in Figure 5c remains physiological and similar to the one used for the training in Figure 5a. Figure 5d shows the rejected (thinned) events of the process (but not their pruned offsprings), whose realization could have potentially yielded explosive rates.

It should be noted that the learned MR-PP exhibits a fuzzy behavior: it is both self-excitatory and after some time self-inhibitory capturing a phenomenon of self-regulation [19] in this way. This fact could justify the choice of the soft relational constraints induced by the Sparse Normal-Gamma prior instead of a hard, Bernoulli-dictated constraint (for capturing a purely excitatory or purely inhibitory effect). Figures 5e-5g present a similar analysis for single-neuron activity from human cortex [35]. Note that the learned model in Figure 5g was simulated for a longer period (80 seconds) than the observation in Figure 5e (10 seconds). We plot only the last 10 seconds. The full simulated spike train for Figure 5g, the learned intensity functions, the values of the hyperparameters and the parameters of the learning algorithm are provided in the Supplementary Material.

### 4.3 Experimental results on multi-neuron spike train data

In this section, we apply the proposed model to a data set consisting of spike train recordings from 25 neurons in the cat primary visual cortex (area 17) under spontaneous activity. The data is publicly available and can be downloaded from the NSF-funded CRCNS data repository [51]. The dataset was acquired with multi-channel silicon electrode arrays that enable simultaneous recording from many single units at once. This is of utmost importance because recordings from multiple neurons at a time are necessary if conclusions about cortical circuit function or network dynamics are to be derived. In Figure 6a, we visualize the spike train used in the experiment. We used the spikes that are contained in the time-window $[0, 13000]$ $msec$ for learning a MR-PP and those in $[13000, 26000]$ $msec$ for testing it. Both the training and the testing spike sequences contain roughly

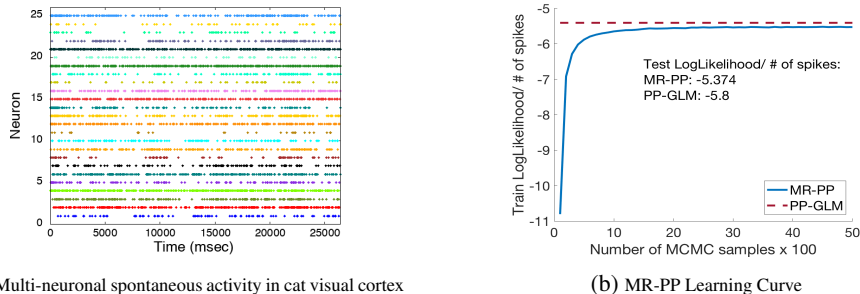

(a) Multi-neuronal spontaneous activity in cat visual cortex

(b) MR-PP Learning Curve

Figure 6: *Multi-neuronal spike train analysis.* 6a visualizes the spike trains for a population of 25 neurons that were used for fitting and testing the multivariate MR-PP. 6b shows the training log-likelihood of the MR-PP with mode point posterior estimates for an increasing number of MCMC batches of 100 samples. The training log-likelihood reaches this of the fitted PP-GLM. However, the log-likelihood for the held-out, second half of the spike train in 6b, is larger for the MR-PP and close to the training log-likelihood.

3,000 spikes each. In Figure 6b, we plot the learning curve (the training data log-likelihood of the spike stream realized with respect to the total number of Markov Chain Monte Carlo (MCMC) samples - the 2000 burn-in samples are also included). The predictive log-likelihood (normalized by the number of spikes) achieved by the posterior mode estimates (from the last 3000 MCMC samples) for the second half of Figure 6a is $-5.374$. We also fit a Poisson-GLM with log link function assuming intensities of the same form as in [33] provided by the statistical python package *StatsModels*. We adjusted the time discretization interval needed to get the spike counts and the order of the regression ($\Delta t = 0.1 \ msec$ and $Q = 1$, respectively), so that the predictive log-likelihood for the spikes in $[13000, 26000]$ is maximized. $StatsModels$ uses Iteratively Reweighted Least Squares (IRLS) for efficiently fitting GLMs. No regularization was incorporated in the model. Assuming that $\Delta t$ is small enough so that there is at most one spike in each one of the $B = T/\Delta t$ time bins in the interval $[0, T]$, the discrete-time log-likelihood of the $J_b$ spike counts in the time bins $T_b$, for $b = 1, 2, \ldots, B$ is given by

$$\log p(J_{1:B}|\boldsymbol{\theta}) = \sum_{b=1}^{B} \log(\lambda(T_b|\boldsymbol{\theta}, H_b)\Delta t)J_b - \sum_{b=1}^{B} \lambda(T_b|\boldsymbol{\theta}, H_b)\Delta t + J \log(\Delta t), \qquad (22)$$

where $J = \sum_{b=1}^{B} J_b$ is the total number of spikes, $\boldsymbol{\theta}$ the Poisson-GLM parameters and $H_b$ the spiking count history in the last $Q$ time bins before the $b$-th time bin. For sufficiently small $\Delta t$, it can be proved [33] that Equation (22) is a discrete time approximation of the continuous time log-likelihood in Equation (12). For fair comparison, in Figure 6b we are subtracting the term $Jlog(\Delta t)$ from the log-likelihood reported by the *StatsModels* (Equation (22)). The hyperparameters, the learning parameters, and the inference time are given in the Supplementary Material.

## 5   Discussion

In this paper, we have presented the first Bayesian, continuous time, point process model which can capture nonlinear, potentially inhibitory, temporal dependencies. A joint prior for the model parameters was designed so that soft relational constraints between types of events are established. The model has managed to recover physiological, single-neuronal dynamics, unlike prevalent alternatives, while still achieving competitive forecasting capacity for multi-neuronal recordings.

There are several avenues for practical utility of the proposed model, such as analyses of physiological mechanisms which are abundant of complex temporal interactions between events of various types and are characterized by relative data scarcity. For example, vital signs monitoring [52, 53], dynamic modeling of biological networks [54, 55] or temporal modeling of clinical events [56], where inhibitory effects may e.g. represent medical therapies or treatments, could be potential application domains of mutually regressive point processes.

There is a multitude of learning tasks that can be augmented with the use of 'signed' relationships, so that they can leverage both the excitatory and the inhibitory interactions that MR-PP can describe such as discovering causality [31] or network structure [57, 45]. Finally, prior sensitivity analysis, a design strategy for hyperparameters selection and development of stochastic variational inference algorithms [44] for large-scale MR-PPs are left for future research.

# 6 Acknowledgments

This work was partially supported by DARPA under award FA8750-17-2-013, in part by the Alexander Onassis Foundation graduate fellowship and in part by the A. G. Leventis Foundation graduate fellowship. We would also like to thank Sibi Venkatesan and Jeremy Cohen for their useful feedback on the paper and Alex Reinhart for helpful discussions.

## Footnotes

[1]The library is written in C++. Our code is available at https://github.com/ifiaposto/Mutually-Regressive-Point-Processes

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
