[Supplementary Material]

# Supplementary Material for the Paper: Mutually Regressive Point Processes

In the following sections, we denote by

$$\{t_1, t_2, \dots\} \sim \mathcal{PP}(\lambda(t)), \tag{S1}$$

a realization of a point process characterized by the intensity function $\lambda(t)$.

## S.I   Correctness Proof of a MR-PP simulation

**Theorem 1** *Assume that a set of events is sampled from a HP $\mathcal{HP}_N(\lambda_n^*(t))$. Afterwards, the simulated events are accepted with probability $\lambda_n(t)/\lambda_n^*(t) = p_n(t)$, where $p_n(t)$ is defined in Equation (5). In case an event is rejected, its offsprings are pruned so that the intensity $\lambda_n^*(t)$ defined in Equation (4) depends only on the realized events whose arrival times are notated as $\dot{t}_i^m$. Let $\dot{\mathcal{S}}_n \triangleq \{\dot{t}_i^n, \dot{z}_i^n\}_{i=1}^{K_n}$ be the sequence of the $K_n$ realized (observed) events of type $n$ augmented with the cluster structure: $\dot{z}_i^n$ is the arrival time of the event which triggered the $i$-th event of type $n$. Similarly, $\tilde{\mathcal{S}}_n \triangleq \{\tilde{t}_i^n, \tilde{z}_i^n\}_{i=1}^{M_n}$ is the sequence of the thinned events of type $n$. Then,*

$$\dot{\mathcal{S}}_n \sim \mathcal{PP}(\lambda_n^*(t) p_n(t)), \text{ and} \tag{S2}$$

$$\tilde{\mathcal{S}}_n \sim \mathcal{PP}(\lambda_n^*(t)(1 - p_n(t))). \tag{S3}$$

**Proof:** Note in the above description that the thinned events are not observed; hence they constitute latent variables of the model. Moreover, the variables $\dot{z}_i^n$ and $\tilde{z}_i^n$ are latent for both the observed events and the thinned events. The proof is similar to the proof in [*S1*]. We assume a temporal order of the observed events such that $0 < \dot{t}_1^n < \dot{t}_2^n < \dot{t}_3^n < \cdots < \dot{t}_{K_n}^n$. Similarly, for the thinned events, $0 < \tilde{t}_1^n < \tilde{t}_2^n < \tilde{t}_3^n < \cdots < \tilde{t}_{M_n}^n$. The merged sequence of the realized and thinned events is denoted by:

$$\mathcal{S}_n \triangleq \{t_i^n, z_i^n, s_i^n,\}_{i=1}^{K_n+M_n}, \tag{S4}$$

such that $0 < t_1^n < t_2^n < t_3^i < \cdots < t_{K_n+M_n}^n$. The variables $s_i^n$ correspond to the label that indicates whether the event will be realized ($s_i^n = 1$) or thinned ($s_i^n = 0$).

For Equation (S2), we will prove that:

$$p(\dot{\mathcal{S}}_n \mid \lambda_n^*(t), p_n(t)) = \exp\left\{-\int_0^T \lambda_n^*(\tau)\, p_n(\tau)\, \mathrm{d}\tau\right\} \times \prod_{i=1}^{K_n} \lambda_n^*(\dot{t}_i^n)\, p_n(\dot{t}_i^n). \tag{S5}$$

Then, Equation (S3) can follow from Bayes rule. From the generative procedure described above, it holds that:

$$p(\mathcal{S}_n \mid \lambda_n^*(t), p_n(t)) =$$
$$\exp\left\{-\int_0^T \lambda_n^*(\tau)\, \mathrm{d}\tau\right\} \times \prod_{i=1}^{K_n+M_n} \lambda_n^*(t_i^n) \times \prod_{i=1}^{M_n+K_n} p_n(t_i^n)^{s_i^n} \left(1 - p_n(t_i^n)\right)^{1-s_i^n}. \tag{S6}$$

The first two terms in Equation (S6) are related to the generation of the arrival times from the Hawkes process. The last one is associated with the labeling of the events and can be derived by the chain

rule according to the chronological order of the events. Note that this likelihood is different from the joint probability $p(t_1^n, t_2^n, \ldots, t_{K_n+M_n}^n)$ since it also involves the probability of the outcome of the thinning for each event. Then, from Bayes rule and Equations (S5), (S6), Equation (S3) follows:

$$p(\tilde{\mathcal{S}}_n \mid \lambda_n^*(t), p_n(t)) = \frac{p(S_n \mid \lambda_n^*(t), p_n(t))}{p(\dot{\mathcal{S}}_n \mid \lambda_n^*(t), p_n(t))} \Rightarrow$$

$$p(\tilde{\mathcal{S}}_n \mid \lambda_n^*(t), p_n(t)) = \exp\left\{ -\int_0^T \lambda_n^*(t)\,(1 - p_n(\tau))\,\mathrm{d}\tau \right\} \times \prod_{i=1}^{M_n} \lambda_n^*(\tilde{t}_i^n)\,(1 - p_n(\tilde{t}_i^n)). \quad \text{(S7)}$$

Assuming a cluster representation [*S2*], Equation (S6) can be rewritten as [*S3*]:

$$p(\mathcal{S}_n \mid \lambda_n^*(t), p_n(t)) = p(\mathcal{S}_n^0 \mid \lambda_n^*(t), p_n(t)) \times \prod_{m=1}^{N} \prod_{j=1}^{K_m} p(\mathcal{S}_n^{\dot{t}_j^m} \mid \lambda_n^*(t), p_n(t)), \quad \text{(S8)}$$

$$p(\mathcal{S}_n^0 \mid \lambda_n^*(t), p_n(t)) = \exp(-\lambda_n^* T) \times \prod_{i=1}^{K_n} (\lambda_n^* p_n(\dot{t}_i^n))^{\mathcal{I}(\dot{z}_i^n=0)} \times \prod_{i=1}^{M_n} (\lambda_n^*(1 - p_n(\tilde{t}_i^n)))^{\mathcal{I}(\tilde{z}_i^n=0)}, \quad \text{(S9)}$$

$$p(\mathcal{S}_n^{\dot{t}_j^m} \mid \lambda_n^*(t), p_n(t)) =$$
$$\exp\left\{ -\int_{\dot{t}_j^m}^T \lambda_{m,n}(\tau, \dot{t}_j^m) d\tau \right\} \times \prod_{i=1}^{K_n} (\lambda_{m,n}(\dot{t}_i^n, \dot{t}_j^m) p_n(\dot{t}_i^n))^{\mathcal{I}(\dot{z}_i^n=\dot{t}_j^m)}$$
$$\times \prod_{i=1}^{M_n} (\lambda_{m,n}(\tilde{t}_i^n, \dot{t}_j^m)(1 - p_n(\tilde{t}_i^n)))^{\mathcal{I}(\tilde{z}_i^n=\dot{t}_j^m)}, \quad \text{(S10)}$$

where $\mathcal{S}_n^0$ are the events with $z_i^n = 0$ that are generated by the exogenous intensity $\lambda_n^*$, and $\mathcal{S}_n^{\dot{t}_j^m}$ the events that are generated by the observed event of type $m$ occurred at $\dot{t}_j^m$. Time 0 is associated with a virtual event which generates events according to the exogenous intensity. $z_i^n$ equals the arrival time of the parent of the event at $t_i^n$. In order to obtain Equation (S5) from Equation (S8), we will marginalize the thinned events. Equivalently, we will marginalize the thinned events in each one of Equations (S9), (S10). We first marginalize over the arrival times of the thinned events:

$$p(\dot{\mathcal{S}}_n^{\dot{t}_j^m}, M_n^{\dot{t}_j^m} \mid \lambda_n^*(t), p_n(t)) =$$
$$\exp\left\{ -\int_{\dot{t}_j^m}^T \lambda_{m,n}(\tau, \dot{t}_j^m) d\tau \right\} \times \prod_{i=1}^{K_n} (\lambda_{m,n}(\dot{t}_i^n, \dot{t}_j^m) p_n(\dot{t}_i^n))^{\mathcal{I}(\dot{z}_i^n=\dot{t}_j^m)}$$
$$\times \frac{(\int_{\dot{t}_j^m}^T \lambda_{m,n}(\tau, \dot{t}_j^m)\,(1 - p_n(\tau)) d\tau)^{M_n^{\dot{t}_j^m}}}{M_n^{\dot{t}_j^m}!}. \quad \text{(S11)}$$

In the above equation, $M_n^{\dot{t}_j^m}$ is the number of thinned events generated by the observed event at $\dot{t}_j^m$. Note that the ordering of $\tilde{t}_i^n$ reduces the integration interval for the marginalization. Subsequently,

we marginalize over the number of the thinned events $M_n^{i_j^m}$:

$$p(\dot{\mathcal{S}}_n^{i_j^m} \mid \lambda_n^*(t), p_n(t)) =$$

$$\exp\left\{-\int_{i_j^m}^T \lambda_{m,n}(\tau, i_j^m)d\tau\right\} \times \prod_{i=1}^{K_n}(\lambda_{m,n}(i_i^n, i_j^m)p_n(i_i^n))^{\mathcal{I}(\dot{z}_i^n = i_j^m)}$$

$$\times \sum_{M_n^{i_j^m}=0}^{\infty} \frac{(\int_{i_j^m}^T \lambda_{m,n}(\tau, i_j^m)(1 - p_n(\tau))d\tau)^{M_n^{i_j^m}}}{M_n^{i_j^m}!} \Rightarrow$$

$$p(\dot{\mathcal{S}}_n^{i_j^m} \mid \lambda_n^*(t), p_n(t)) =$$

$$\exp\left\{-\int_{i_j^m}^T \lambda_{m,n}(\tau, i_j^m)d\tau\right\} \times \prod_{i=1}^{K_n}(\lambda_{m,n}(i_i^n, i_j^m)p_n(i_i^n))^{\mathcal{I}(\dot{z}_i^n = i_j^m)}$$

$$\times \exp\left\{\int_{i_j^m}^T \lambda_{m,n}(\tau, i_j^m)(1 - p_n(\tau))d\tau\right\} \Rightarrow$$

$$p(\dot{\mathcal{S}}_n^{i_j^m} \mid \lambda_n^*(t), p_n(t)) = \exp\left\{-\int_{i_j^m}^T \lambda_{m,n}(\tau, i_j^m)p_n(\tau)d\tau\right\} \times \prod_{i=1}^{K_n}(\lambda_{m,n}(i_i^n, i_m^j)p_n(i_i^n))^{\mathcal{I}(\dot{z}_i^n = i_j^m)}.$$

$$(S12)$$

An identical analysis holds for $p(\dot{\mathcal{S}}_n^0 \mid \lambda_n^*(t), p_n(t))$. By multiplying Equations (S12), for $m = 1, 2, \ldots, N$ and $j = 1, 2, \ldots, K_m$ (assuming a cluster representation similar to that in Equation (S8) for the observed events), Equation (S5) is obtained.

## S.II  Details of the Bayesian Inference Algorithm

In Section S.II.A, we provide the computation of the likelihood that will be used for the derivations of the posterior distributions and the Metropolis-Hastings ratios of the inference algorithm. In Section S.II.B, we provide the Gibbs updates for the parameters in the intensity function $\lambda_n^*(t)$. In Section S.II.C, we provide the derivations for the conjugate Gibbs update of the interaction weights. In Section S.II.D, we derive the collapsed Metropolis-Hastings ratio for the update of the mean and precision of the Gaussian prior for the interaction weights. Finally, in Section S.II.E, we derive the updates for the parameters of the history kernel function.

The model parameters related to the effect on the point process of events of type $n$ are represented as vectors. Specifically, the excitatory and decaying coefficients in the intensity function in Equations (4),(2) are represented as:

$$\boldsymbol{\alpha}_n = [\alpha_{1,n}, \alpha_{2,n}, \ldots, \alpha_{N,n}], \text{and } \boldsymbol{\delta}_n = [\delta_{1,n}, \delta_{1,n}, \ldots, \delta_{N,n}]. \tag{S13}$$

Similarly, the weights, the precision and the mean of their priors are represented as:

$$\boldsymbol{w}_n = [b_n, w_{1,n}, w_{2,n}, \ldots, w_{N,n}]^T, \tag{S14}$$

$$\boldsymbol{\tau}_n = [1/\sigma_0^2, \tau_{1,n}, \tau_{2,n}, \ldots, \tau_{N,n}]^T, \tag{S15}$$

$$\boldsymbol{\mu}_n = [\mu_0, \mu_{1,n}, \mu_{2,n}, \ldots, \mu_{N,n}]^T, \tag{S16}$$

$$\boldsymbol{\Sigma}_n = diag(\boldsymbol{\tau}_n)^{-1}. \tag{S17}$$

We assume the following priors, unless otherwise mentioned,:

$$\lambda_n^* \sim \text{Gamma}(\alpha_e, \beta_e), \tag{S18}$$

$$\alpha_{m,n} \sim \text{Gamma}(\alpha_m, \beta_m), \tag{S19}$$

$$\delta_{m,n} \sim \text{Exp}(\lambda_\delta), \tag{S20}$$

$$c \sim \text{Gamma}(\alpha_c, \beta_c), \tag{S21}$$

$$\gamma \sim \text{Exp}(\lambda_\gamma), \tag{S22}$$

where $\alpha_e > 0$, $\beta_e > 0$, $\alpha_m > 0$, $\beta_m > 0$, $\lambda_\delta > 0$, $\alpha_c > 0$, $\beta_c > 0$, and $\lambda_\gamma > 0$. We denote by $p(\alpha)$ the probability density of the prior of the scalar parameter $\alpha$.

Let $\boldsymbol{\theta}$ be the set of the hyperparameters in Equations (S18)-(S22), and (8)-(11):

$$\boldsymbol{\theta} \triangleq \{\alpha_e, \beta_e, \alpha_m, \beta_m, \lambda_\delta, \alpha_c, \beta_c, \lambda_\gamma, \mu_0, \sigma_0^2, \nu_\tau, \alpha_\tau, \beta_\tau, \nu_\mu, \alpha_\mu, \lambda_\mu, \alpha_0, \delta_0\}. \tag{S23}$$

## S.II.A   Derivation of the data likelihood

The likelihood of the sequence $\mathcal{T}_n \triangleq \{t_i^n\}_{i=1}^{K_n}$ of $K_n$ events generated by a point process with intensity function $\lambda_n(t)$ in the time window $[0, T]$ is [S4]:

$$p(\mathcal{T}_n \mid \lambda_n(t)) = \exp\left\{-\int_0^T \lambda_n(t)\,\mathrm{d}t\right\} \prod_{i=1}^{K_n} \lambda_n(t_i^n). \tag{S24}$$

The likelihood in Equation (S24) will be sequentially augmented with the following latent variables in order to derive conjugate updates for $\lambda_n^*$, $\alpha_{m,n}$ (for the case of a flat MR-PP), and $\boldsymbol{w}_n$. These latent variables are:

1. parent variables $z_i^n = \dot{t}_j^m$ for the $i$-th event of type $n$ occurred at time $t_i^n$, in the sense that it belongs to the Poisson process $\lambda_{m,n}(t, \dot{t}_j^m)$ generated by the $j$-th observed event of type $m$ occurred at time $\dot{t}_j^m$.

2. $M_n$ thinned events $\tilde{t}_i^n$ of type $n$.

3. Pólya-Gamma random variables $\omega_i^n$ associated with an event at $t_i^n$.

Each one of the following subsections provides the likelihood for the inclusion of the latent variables described above.

### S.II.A.1   Likelihood of a cluster MR-PP with thinned events

The likelihood of the augmented sequence $\mathcal{S}_n$ according to Equation (S4) given the model parameters is:

$$p(\mathcal{S}_n \mid \lambda_n^*, \boldsymbol{\alpha}_n, \boldsymbol{\delta}_n, \boldsymbol{w}_n, c, \gamma) =$$
$$\exp(-\lambda_n^* T) \times \prod_{m=1}^{N} \prod_{j=1}^{K_m} \exp\left\{-\int_{\dot{t}_j^m}^{T} \alpha_{m,n} e^{-\delta_{m,n}(\tau - \dot{t}_j^m)} d\tau\right\} \times$$
$$\prod_{i=1}^{K_n + M_n} [\lambda_n^{*\mathcal{I}(z_i^n = 0)} \times \prod_{m=1}^{N} \prod_{j=1}^{K_m} (\alpha_{m,n} e^{-\delta_{m,n}(t_i^n - \dot{t}_j^m)})^{\mathcal{I}(z_i^n = \dot{t}_j^m)}] \times$$
$$\prod_{i=1}^{K_n + M_n} \frac{e^{(\boldsymbol{w}_n^T \boldsymbol{h}(t_i^n)) \times s_i^n}}{e^{\boldsymbol{w}_n^T \boldsymbol{h}(t_i^n)} + 1}. \tag{S25}$$

Intuitively, Equation (S25) can be viewed as the likelihood of two constituent procedures: a point process $\mathcal{PP}(\lambda_n^*(t))$, which generates the arrival times $\mathcal{T}_n$, and a thinning procedure which generates the labels of the events. Note that there is no circularity since the probability $p_n(t)$ and the intensity $\lambda_n^*(t)$ depend on the realized events which occurred before time $t$. Therefore, the likelihood in Equation (S25) is well defined. The first term in Equation (S25) stems from the fact that there are $1 + \sum_{m=1}^{N} K_m$ processes which generate events of type $n$: the exogenous, homogeneous Poisson $\mathcal{PP}(\lambda_n^*)$ and the non-homogeneous Poisson processes generated by the $K_m$ realized events of type $m$. Each one of them contributes an exponential term to the likelihood according to Equation (S24). The second term is related to the assignment of the events to their parents, so that the product in Equation (S25) contains only the events that belong to the point process trigerred by the event $\dot{t}_j^m$. The last term in Equation (S25) accounts for the label of the events (whether the event is realized or not). Note that the dependence on the realized events of the rest of the types is implicitly assumed in the aforementioned formulas.

### S.II.A.2 Pólya-Gamma augmented likelihood of a cluster MR-PP with thinned events

We define the likelihood contribution of the thinning acceptance/ rejection of an event at time $t_i^n$ as:

$$\ell_i^n \triangleq \frac{e^{(\boldsymbol{w}_n^T \boldsymbol{h}(t_i^n)) \times s_i^n}}{e^{\boldsymbol{w}_n^T \boldsymbol{h}(t_i^n)} + 1}. \tag{S26}$$

According to Theorem 1 in [S5], it can be rewritten as:

$$\ell_i^n \propto \exp(\nu_i^n \boldsymbol{w}_n{}^T \boldsymbol{h}(t_i^n)) \int_0^\infty \exp\left\{-\frac{1}{2}\omega_i^n(\boldsymbol{w}_n^T \boldsymbol{h}(t_i^n))^2\right\} \mathcal{PG}_m(\omega_i^n; 1, 0)\, d\omega_i^n, \tag{S27}$$

where $\nu_i^n = s_i^n - 1/2$, and $\mathcal{PG}_m(\omega_i^n; 1, 0)$ is the density of a Pólya-Gamma distribution with parameters $(1, 0)$. Combined with a prior on $\boldsymbol{w}_n$, the integrand in Equation (S27) defines a joint density on $(s_i^n, \omega_i^n, \boldsymbol{w}_n)$, where $\omega_i^n$ is a latent Pólya-Gamma random variable. We associate a Pólya-Gamma random variable with each event, and the event sequences become: $\dot{SP}_n \triangleq \{\dot{t}_i^n, \dot{z}_i^n, \dot{\omega}_i^n\}_{i=1}^{K_n}$, $\tilde{\mathcal{SP}}_n \triangleq \{\tilde{t}_i^n, \tilde{z}_i^n, \tilde{\omega}_i^n\}_{i=1}^{M_n}$, $\mathcal{SP}_n \triangleq \{t_i^n, s_i^n, z_i^n, \omega_i^n\}_{i=1}^{K_n+M_n}$. Therefore, Equation (S25) becomes:

$$p(\mathcal{SP}_n \mid \lambda_n^*, \boldsymbol{\alpha}_n, \boldsymbol{\delta}_n, \boldsymbol{w}_n, c, \gamma) \propto$$

$$\exp(-\lambda_n^* T) \times \prod_{m=1}^{N} \prod_{j=1}^{K_m} \exp\left\{-\int_{\dot{t}_j^m}^T \alpha_{m,n} e^{-\delta_{m,n}(\tau - \dot{t}_j^m)} d\tau\right\} \times$$

$$\prod_{i=1}^{K_n+M_n} [\lambda_n^{* \mathcal{I}(z_i^n=0)} \times \prod_{m=1}^{N} \prod_{j=1}^{K_m} (\alpha_{m,n} e^{-\delta_{m,n}(t_i^n - \dot{t}_j^m)})^{\mathcal{I}(z_i^n = \dot{t}_j^m)}] \times$$

$$\prod_{i=1}^{K_n+M_n} \exp\left\{\nu_i^n \boldsymbol{w}_n{}^T \boldsymbol{h}(t_i^n) - \tfrac{1}{2}\omega_i^n(\boldsymbol{w}_n^T \boldsymbol{h}(t_i^n))^2\right\} \mathcal{PG}_m(\omega_i^n; 1, 0). \tag{S28}$$

### S.II.B Gibbs updates for the intensity functions

#### S.II.B.1 Gibbs update for the exogenous rates $\lambda_n^*$

Given $\mathcal{S}_n$ as defined in Equation (S4), conjugate updates are possible for the intensity parameters $\lambda_n^*$ [S6].

From Equations (S18), (S28), and by keeping the terms in which $\lambda_n^*$ appears, we obtain:

$$p(\lambda_n^* \mid \mathcal{S}_n) \propto \lambda_n^{* N_n^*} \exp(-\lambda_n^* T)\, \mathrm{Gamma}(\lambda_n^*; \alpha_e, \beta_e), \tag{S29}$$

where $N_n^*$ is the number of events of type $n$ which belong to the exogenous process $\lambda_n^*$. Therefore,

$$p(\lambda_n^* \mid \mathcal{S}_n) = \mathrm{Gamma}(\lambda_n^*; \tilde{\alpha}_e^n, \tilde{\beta}_e^n), \tag{S30}$$

$$\tilde{\alpha}_e^n = N_n^* + \alpha_e, \tag{S31}$$

$$\tilde{\beta}_e^n = T + \beta_e. \tag{S32}$$

#### S.II.B.2 Gibbs update for the endogenous rates $\alpha_{m,n}$

Given $\mathcal{S}_n$ and $\dot{\mathcal{T}}_m$, conjugate prior updates are possible for the endogenous intensity parameters $\alpha_{m,n}$ for a flat MR-PP [S6] and a Gamma prior. This is due to the independence between the parameters in the thinning portion $p_n(t)$ and the intensity-related portion $\lambda_n^*(t)$ of the model.

From Equations (S19), (S28), the update for the mutually excitatory coefficients $\alpha_{m,n}$ is given by:

$$p(\alpha_{m,n} \mid \{z_i^n\}_{i=1}^{K_n+M_n}, \{\dot{t}_i^m\}_{i=1}^{K_m}, \delta_{m,n}) \propto \mathcal{L}_{\alpha_{m,n}} \times p(\alpha_{m,n}), \tag{S33}$$

where $\mathcal{L}_{\alpha_{m,n}}$ the factors of the likelihood in (S28) that include $\alpha_{m,n}$:

$$\mathcal{L}_{\alpha_{m,n}} = exp\left\{-\alpha_{m,n} \sum_{i=1}^{K_m} \int_{\dot{t}_i^m}^T exp\left(-\delta_{m,n}(\tau - \dot{t}_i^m)\right) d\tau\right\} \alpha_{m,n}^{N_{m,n}}$$

$$= exp\left\{-\alpha_{m,n}\left(\frac{K_m}{\delta_{m,n}} - \frac{1}{\delta_{m,n}} \sum_{i=1}^{K_m} exp(-\delta_{m,n}(T - \dot{t}_i^m))\right)\right\} \alpha_{m,n}^{N_{m,n}}, \tag{S34}$$

and $N_{m,n}$ is the number of events of type $n$ triggered by an event of type $m$. Finally, if a gamma prior is assumed, i.e. $p(\alpha_{m,n}) = \text{Gamma}(\alpha_{m,n}; \alpha_m, \beta_m)$,

$$p(\alpha_{m,n} \mid \dots) = \text{Gamma}(\alpha_{m,n}; \tilde{\alpha}_m^{m,n}, \tilde{\beta}_m^{m,n}), \tag{S35}$$

$$\tilde{\alpha}_m^{m,n} = \alpha_m + N_{m,n}, \tag{S36}$$

$$\tilde{\beta}_m^{m,n} = \beta_m + \frac{1}{\delta_{m,n}} \big( K_m - \sum_{i=1}^{K_m} exp(-\delta_{m,n}(T - \dot{t}_i^m)) \big). \tag{S37}$$

In case of a hierarchical MR-PP, $\alpha_{m,n}$ will be updated jointly with the hyperparameters of the corresponding weight, see Subsection S.II.D.

### S.II.B.3   Gibbs sampling for the events' cluster structure

The parent of each event follows a categorical posterior distribution. Let $\dot{z}_i^n$ be the arrival time of the observed event which triggered the $i$-th observed event of type $n$. The set of potential parent events consists of the observed events (of the same or different type) that occurred before $\dot{t}_i^n$ and it is denoted by

$$\mathbb{Z}_i^n \triangleq \big\{ \{ \dot{t}_j^m : \dot{t}_j^m < \dot{t}_i^n \}_{j=1}^{K_m} \big\}_{m=1}^{N}. \tag{S38}$$

Let $\mathbb{P}_i^n \triangleq \big\{ p(\dot{z}_i^n = \dot{t} \mid \dots) \big\}_{\dot{t} \in \mathbb{Z}_i^n}$ be the corresponding posterior selection probabilities. From Equation (S28), we can obtain:

$$p(\dot{z}_i^n = 0 \mid \dots) = \frac{\lambda_n^*}{\mathcal{Z}}, \tag{S39}$$

$$p(\dot{z}_i^n = \dot{t}_j^m \mid \dots) = \frac{\alpha_{m,n} e^{-\delta_{m,n}(\dot{t}_i^n - \dot{t}_j^m)}}{\mathcal{Z}}, \tag{S40}$$

$$\mathcal{Z} = \lambda_n^* + \sum_{m=1}^{N} \sum_{j=1}^{K_m} \alpha_{m,n} e^{-\delta_{m,n}(\dot{t}_i^n - \dot{t}_j^m)} \mathcal{I}(\dot{t}_j^m < \dot{t}_i^n). \tag{S41}$$

Although there are $\sum_{n=1}^{N} K_n$ parent variables, they are conditionally independent and may be sampled in parallel.

### S.II.B.4   Metropolis update for the decaying coefficients $\delta_{m,n}$

A Gaussian proposal distribution that is updated according to the adaptive Metropolis algorithm [S7] is used for $\delta_{m,n}$.

Let $\delta_{m,n}'$ be the proposed value of $\delta_{m,n}$. From Equations (S20), (S28), the Metropolis ratio $H_{\delta_{m,n}}$ will be:

$$H_{\delta_{m,n}} = H_{\delta_{m,n}}^1 \times H_{\delta_{m,n}}^2 \times H_{\delta_{m,n}}^3, \tag{S42}$$

$$H_{\delta_{m,n}}^1 = \exp\{ -\lambda_\delta (\delta_{m,n}' - \delta_{m,n}) \}, \tag{S43}$$

$$H_{\delta_{m,n}}^2 = \exp\left\{ (\delta_{m,n} - \delta_{m,n}') \times \sum_{j=1}^{K_m} \sum_{i=1}^{K_n + M_n} (t_i^n - \dot{t}_i^m) \mathcal{I}(z_i^n = \dot{t}_j^m) \right\}, \tag{S44}$$

$$H_{\delta_{m,n}}^3 =$$

$$\exp\left\{ \alpha_{m,n} \left[ K_m \left( \frac{1}{\delta_{m,n}} - \frac{1}{\delta_{m,n}'} \right) + \sum_{i=1}^{K_m} \left( \frac{\exp(-\delta_{m,n}'(T - \dot{t}_i^m))}{\delta_{m,n}'} - \frac{\exp(-\delta_{m,n}(T - \dot{t}_i^m))}{\delta_{m,n}} \right) \right] \right\} \tag{S45}$$

## S.II.C   Gibbs Sampling for the interaction weights

For obtaining the posterior of the interaction weights, we keep from Equations (10), (S28) only the terms which contain $\boldsymbol{w}_n$:

$$p(\boldsymbol{w}_n \mid \dots) \propto$$

$$\mathcal{N}(\boldsymbol{w}_n; \boldsymbol{\mu}_n, \boldsymbol{\Sigma}_n) \times \prod_{i=1}^{K_n+M_n} \exp\left\{\nu_n^i \boldsymbol{w}_n^T \boldsymbol{h}(t_i^n) - \tfrac{1}{2}\omega_n^i (\boldsymbol{w}_n^T \boldsymbol{h}(t_i^n))^2\right\} \propto$$

$$\mathcal{N}(\boldsymbol{w}_n; \boldsymbol{\mu}_n, \boldsymbol{\Sigma}_n) \times \prod_{i=1}^{K_n+M_n} \exp\left\{-\frac{\omega_i^n}{2}(\boldsymbol{w}_n^\mathsf{T} \boldsymbol{h}(t_i^n) - \nu_i^n/\omega_i^n)^2\right\} \propto$$

$$\exp\left\{-\frac{1}{2}(\boldsymbol{w}_n - \boldsymbol{\mu}_n)^T \boldsymbol{\Sigma}_n^{-1}(\boldsymbol{w}_n - \boldsymbol{\mu}_n)\right\} \times \exp\left\{-\frac{1}{2}(\boldsymbol{z}_n - \boldsymbol{H}_n \boldsymbol{w}_n)^T \boldsymbol{\Omega}_n (\boldsymbol{z}_n - \boldsymbol{H}_n \boldsymbol{w}_n)\right\},$$
$$\text{(S46)}$$

where:

$$\boldsymbol{\Omega}_n = \operatorname{diag}(\omega_1^n, \omega_2^n, \dots, \omega_{K_n+M_n}^n), \tag{S47}$$

$$\boldsymbol{z}_n = [\nu_1^n/\omega_1^n, \dots, \nu_{K_n+M_n}^n/\omega_{K_n+M_n}^n]^T, \tag{S48}$$

$$\boldsymbol{H}_n = [\boldsymbol{h}(t_1^n), \dots, \boldsymbol{h}(t_{K_n+M_n}^n)]^T. \tag{S49}$$

Finally,

$$p(\boldsymbol{w}_n \mid \dots) = \mathcal{N}(\boldsymbol{w}_n; \tilde{\boldsymbol{\Sigma}}_n, \tilde{\boldsymbol{\mu}}_n). \tag{S50}$$

By equating the quadratic and linear terms of $\boldsymbol{w}_n$, we get $\tilde{\boldsymbol{\Sigma}}_n$ and $\tilde{\boldsymbol{\mu}}_n$ respectively:

$$\tilde{\boldsymbol{\Sigma}}_n = \left(\boldsymbol{\Sigma}_n^{-1} + \boldsymbol{H}_n^T \boldsymbol{\Omega}_n \boldsymbol{H}_n\right)^{-1}, \tag{S51}$$

$$\tilde{\boldsymbol{\mu}}_n = \tilde{\boldsymbol{\Sigma}}_n \left(\boldsymbol{\Sigma}_n^{-1} \boldsymbol{\mu}_n + \boldsymbol{H}_n^T \boldsymbol{\Omega}_n \boldsymbol{z}_n\right). \tag{S52}$$

Note that the labels of the events contribute to the new sample of the weight through the terms $\nu_i^n$. The sign of these terms (positive for a realized event and negative for a thinned event) steers the new sample of the weight to either a positive or a negative value according to Equation (S52). The contribution of these terms is weighted by the history of the corresponding event. In case of negligible history the second terms of the summation in the updates of Equations (S51) and (S52) have small effect.

From Theorem 1 in [S5], for $\alpha = 1$ and $\beta = 1$, the posterior for sampling $\omega_i^n$ is

$$p(\omega_i^n \mid \dots) = p(\omega_i^n \mid \{\dot{\mathcal{T}}_{n'}\}_{n'=1}^N, \boldsymbol{w}_n, c, \gamma) = \mathcal{PG}_m(\omega_i^n; 1, \boldsymbol{w}_n^T \boldsymbol{h}(t_n^i)). \tag{S53}$$

Although there are $\sum_{n=1}^N K_n + M_n$ Pólya-Gamma random variables that have to be sampled, they are independent and can be sampled in parallel.

## S.II.D   Collapsed Metropolis-Hastings for the weights' prior mean $\mu_{m,n}$ and precision $\tau_{m,n}$ and the endogenous excitation rate $\alpha_{m,n}$

In case of a hierarchical MR-PP, $\alpha_{m,n}$ is coupled with $\mu_{m,n}$, and $\tau_{m,n}$ through the Equations (8)-(10). Therefore, the conjugate update of Equations (S35)-(S37) is no longer available and $\alpha_{m,n}$, $\mu_{m,n}$ and $\tau_{m,n}$ will be jointly updated with a Metropolis-Hastings step. From Equations (S19), (8), (9), (10), (S28), and by keeping only the terms in the likelihood that include $\mu_{m,n}, \tau_{m,n}, \alpha_{m,n}, \boldsymbol{w}_n$, we will first find the joint posterior up to a normalization constant:

$$p(\alpha_{m,n}, \mu_{m,n}, \tau_{m,n}, \boldsymbol{w}_n \mid \dots) \propto$$
$$p(\alpha_{m,n}) \times p(\tau_{m,n} \mid \alpha_{m,n}) \times p(\mu_{m,n} \mid \alpha_{m,n}, \tau_{m,n}) \times p(\boldsymbol{w}_n \mid \boldsymbol{\mu}_n, \boldsymbol{\tau}_n; \boldsymbol{\theta}) \times$$
$$p(\mathcal{SP}_n \mid \lambda_n^*, \boldsymbol{\alpha}_n, \boldsymbol{\delta}_n, \boldsymbol{w}_n, c, \gamma). \tag{S54}$$

The collapsed, with respect to the interaction weights $\boldsymbol{w}_n$, joint posterior will be:

$$p(\alpha_{m,n}, \mu_{m,n}, \tau_{m,n} \mid \dots) = \int p(\alpha_{m,n}, \mu_{m,n}, \tau_{m,n}, \boldsymbol{w}_n \mid \dots) d\boldsymbol{w}_n. \tag{S55}$$

From Equations (S46), (S50), (S54), (S34), (S55), by keeping only the terms which depend on $\boldsymbol{w}_n$ inside the integration and by making use of the normalization constant of the multivariate Gaussian distribution, we obtain:

$$p(\alpha_{m,n}, \mu_{m,n}, \tau_{m,n} \mid \dots) \propto$$

$$\frac{p(\alpha_{m,n}) \times p(\tau_{m,n} \mid \alpha_{m,n}) \times p(\mu_{m,n} \mid \alpha_{m,n}, \tau_{m,n}) \times \mathcal{L}_{\alpha_{m,n}} \times \tau_{m,n}^{1/2} \times \exp\{-\frac{\mu_{m,n}^2 \tau_{m,n}}{2}\}}{|\tilde{\Sigma}_n|^{-1/2} \exp\left\{ -\frac{1}{2}\tilde{\mu}_n^T \tilde{\Sigma}_n^{-1} \tilde{\mu}_n \right\}}.$$

$$(S56)$$

Equation (S56) is used to compute the Metropolis-Hastings ratio. As proposal for $\alpha_{m,n}$ we can use the prior in which case $p(\alpha_{m,n})$ is removed from the ratio since it is cancelled by the probability density of the proposal. Similarly, the Normal Gamma defined in Equations (8), (9) for the current sample $\alpha_{m,n}$ is used as proposal distribution for $\mu_{m,n}$, and $\tau_{m,n}$, hence the second and third terms in the numerator of Equation (S56) are omitted in the ratio since they are cancelled by the probability density of the proposal.

### S.II.E Metropolis updates for the history kernel functions

A Gaussian proposal distribution that is updated according to the adaptive Metropolis algorithm [S7] is used for both $c$ and $\gamma$.

Recall that the aggregated temporal history of the events of type $m$ at time $t$ is defined as:

$$h_m(t) = c \sum_{i=1}^{K_m} e^{-\gamma(t-\dot{t}_i^m)} \mathcal{I}(\dot{t}_m^i < t), \tag{S57}$$

$$\boldsymbol{h}(t) = [1, h_1(t), h_2(t), \dots, h_N(t)]^T. \tag{S58}$$

We also define $\boldsymbol{\hbar}(t)$, as

$$\hbar_m(t) = \sum_{i=1}^{K_m} e^{-\gamma(t-\dot{t}_i^m)} \mathcal{I}(\dot{t}_m^i < t), \tag{S59}$$

$$\boldsymbol{\hbar}(t) = [1, \hbar_1(t), \hbar_2(t), \dots, \hbar_N(t)]^T. \tag{S60}$$

Let $c'$ be the proposed value of c. From Equations (S21), (S28), the Metropolis ratio $H_c$ will be:

$$H_c = H_c^1 \times H_c^2, \tag{S61}$$

$$H_c^1 = \left(\frac{c'}{c}\right)^{\alpha_c - 1} \times \exp\{-\beta_c(c' - c)\}, \tag{S62}$$

$$H_c^2 = \exp\left\{ \Delta c \sum_{n=1}^{N} \sum_{i=1}^{K_n+M_n} \nu_i^n \boldsymbol{w}_n^T \boldsymbol{\hbar}(t_i^n) - \Delta_c^2 \sum_{n=1}^{N} \sum_{i=1}^{K_n+M_n} \frac{1}{2}\omega_i^n (\boldsymbol{w}_n^T \boldsymbol{\hbar}(t_i^n))^2 \right\}, \tag{S63}$$

$$\Delta c \triangleq c' - c, \tag{S64}$$

$$\Delta_c^2 \triangleq c'^2 - c^2. \tag{S65}$$

Let $\gamma'$ be the proposed value of $\gamma$, and $\hbar'_m(t)$, $\boldsymbol{\hbar}'(t)$ the corresponding history kernel functions. From Equations (S22), (S28), the Metropolis ratio $H_\gamma$ will be:

$$H_\gamma = H_\gamma^1 \times H_\gamma^2, \tag{S66}$$

$$H_\gamma^1 = \exp\{-\lambda_\gamma(\gamma' - \gamma)\}, \tag{S67}$$

$$H_\gamma^2 = \exp\left\{ c \sum_{n=1}^{N} \sum_{i=1}^{K_n+M_n} \nu_i^n \boldsymbol{w}_n^T \Delta\boldsymbol{\hbar}(t_i^n) - c^2 \sum_{n=1}^{N} \sum_{i=1}^{K_n+M_n} \frac{1}{2}\omega_i^n [(\boldsymbol{w}_n^T \boldsymbol{\hbar}'(t_i^n))^2 - (\boldsymbol{w}_n^T \boldsymbol{\hbar}(t_i^n))^2] \right\}, \tag{S68}$$

$$\Delta\boldsymbol{\hbar}(t_i^n) \triangleq \boldsymbol{\hbar}'(t_i^n) - \boldsymbol{\hbar}(t_i^n). \tag{S69}$$

## S.III   Experimental Details

Table S1 shows the learning parameters related to the Adaptive Metropolis algorithm [*S7*] that were used in all of the experiments. The same notation as in [*S7*] is adopted.

Table S1: Adaptive Metropolis Parameters

| AM Parameter | Model Parameter | | |
|---|---|---|---|
| | $\delta_{m,n}$ | $c$ | $\gamma$ |
| $\epsilon$ | 0.0001 | 0.0001 | 0.0001 |
| $c_0$ | 0.001 | 0.1 | 0.01 |
| $s_d$ | 10 | 10 | 10 |

In the next subsections, we elaborate on the hyperparameters used in the experiments. The hyperparameters were chosen manually so that the predictive log-likelihood on held-out data is maximized. For the synthetic experiment, the hyperparameters of the true and the learned MR-PP are the same. We also provide some additional experimental results.

## S.III.A  Details of the experimental results on the synthetic validation

Table S2 shows the hyperparameters used in the synthetic experiment. For demonstration purposes, we assumed an exponential prior for $\alpha_{m,n}$, i.e. $\alpha_{m,n} \sim \text{Exp}(\lambda_m)$.

Note that large values for $\delta_0 = 1e5$ in the activation functions $\phi_\tau(\alpha)$ and $\phi_\mu(\alpha)$ indicate hard relational constraints so that an effect from a type $m$ on a type $n$ is either inhibitory (when $\alpha_{m,n} < \alpha_0$) or excitatory (when $\alpha_{m,n} > \alpha_0$). Moreover, the small value for $\alpha_\mu$ indicates a strong inhibitory relationship in case the excitation is below the threshold value $\alpha_0 = 0.015$ since the mean $\mu_{m,n}$ will be sampled from a Gaussian distribution with a large, negative mean. On the other hand, the large values for $\nu_\mu = 1000$ and $\lambda_\mu = 100$ indicate a vanishing inhibitory relationship when the excitation is above the threshold value $\alpha_0$. This is because, in this case, $\mu_{m,n}$ will be sampled from a Gaussian distribution with almost zero mean and variance.

Specifically, when $\alpha_{m,n} > \alpha_0$, $\phi_\tau(\alpha_{m,n}) \approx 1$. Subsequently, the precision $\tau_{m,n}$ is sampled approximately from $\text{Gamma}(\nu_\tau, \beta_\tau)$. Since $\nu_\tau > \beta_\tau$ and in combination with the fact that the mean will be sampled from a Gaussian with almost zero mean and variance, the corresponding weight $w_{m,n}$ will take a value close to zero with high probability. A symmetric scenario holds when $\alpha_{m,n} < \alpha_0$. In this case, $\phi_\tau(\alpha_{m,n}) \approx 0$, and the precision will be sampled approximately from $\text{Gamma}(\alpha_\tau, \beta_\tau)$. Therefore, $\tau_{m,n}$ will take a small value with high probability, and inhibitory relationships are enabled from type $m$ on type $n$.

Table S2: Hyperparameters for the Synthetic Experiment

| Function | Parameter | Value |
|---|---|---|
| | $\alpha_e$ | 7 |
| | $\beta_e$ | 1000 |
| $\lambda_n(t)$ | $\lambda_m$ | 10 |
| | $\lambda_\delta$ | 20 |
| | $\alpha_c$ | 100 |
| $h_m(t)$ | $\beta_c$ | 10 |
| | $\lambda_\gamma$ | 5 |
| | $\mu_0$ | 10 |
| | $\sigma_0$ | 1 |
| | $\nu_\tau$ | 1000 |
| | $\alpha_\tau$ | 10 |
| $\boldsymbol{w}_n$ | $\beta_\tau$ | 1 |
| | $\nu_\mu$ | 1000 |
| | $\alpha_\mu$ | 0.01 |
| | $\lambda_\mu$ | 100 |
| $\phi_\mu(\alpha)$ | $\alpha_0$ | 0.015 |
| | $\delta_0$ | 1e5 |
| $\phi_\tau(\alpha)$ | $\alpha_0$ | 0.015 |
| | $\delta_0$ | 1e5 |

In Figures S1, S2, we plot the autocorrelation coefficients for the endogenous rates $\alpha_{m,n}$ and the interaction weights $\boldsymbol{w}_n$ for $n = 1$ and $n = 2$ respectively to demonstrate the convergence of the MCMC.

(a) Excitation from Type I

(b) Inhibition from Type I

(c) Excitation from Type II

(d) Inhibition from Type II

Figure S1: Autocorrelation coefficient for the parameters of the effect on events of Type I.

(a) Excitation from Type I

(b) Inhibition from Type I

(c) Excitation from Type II

(d) Inhibition from Type II

Figure S2: Autocorrelation coefficient for the parameters of the effect on events of Type II.

### S.III.B  Details of the experimental results on the stability of single neuron spiking dynamics

#### S.III.B.1  Monkey cortex, single neuron dynamics

Table S3 provides the hyperparameters used for the dynamics recovery of the spike trains of the single, monkey cortex neuron. Note that the small value $\delta_0 = 1$ allows for the existence of both a self-exciting and a self-inhibiting behavior. The large value for $\lambda_\delta = 10000$ indicates a slowly decaying self-exciting kernel. On the other hand, the small value for $\lambda_\gamma = 10$ indicates a fast decaying history kernel. These facts imply an initially self-exciting behavior followed by a self-exhaustion phenomenon when the aggregated history becomes large enough so that the thinning procedure has considerable effect on the intensity.

Table S3: Hyperparameters for the Monkey Cortex, Single-Neuronal Experiment

| Function | Parameter | Value |
|---|---|---|
| $\lambda_n(t)$ | $\alpha_e$ | 100 |
| | $\beta_e$ | 10 |
| | $\alpha_m$ | 1 |
| | $\beta_m$ | 1 |
| | $\lambda_\delta$ | 10000 |
| $h(t)$ | $\alpha_c$ | 10 |
| | $\beta_c$ | 1 |
| | $\lambda_\gamma$ | 10 |
| $\boldsymbol{w}_n$ | $\mu_0$ | 0 |
| | $\sigma_0$ | 10 |
| | $\nu_\tau$ | 1 |
| | $\alpha_\tau$ | 10 |
| | $\beta_\tau$ | 1 |
| | $\nu_\mu$ | 10 |
| | $\alpha_\mu$ | 0.01 |
| | $\lambda_\mu$ | 10 |
| $\phi_\mu(\alpha)$ | $\alpha_0$ | 10 |
| | $\delta_0$ | 1 |
| $\phi_\tau(\alpha)$ | $\alpha_0$ | 10 |
| | $\delta_0$ | 1 |

These patterns are further corroborated by Figure S3. The slowly decaying self-exciting kernel results in a step-like function $\lambda^*(t)$. However, once the history kernel $h(t)$ obtains a value high enough due to the occurrence of the past spikes, it starts having a considerable effect on the MR-PP intensity $\lambda(t)$ and steers it to a small value. The thinning procedure prevents $\lambda(t)$ from diverging to explosive firing rates and therefore generating non-physiological patterns.

(a) Observation 1

(b) Observation 2

(c) Observation 3

(d) Observation 4

(e) Observation 6

(f) Observation 6

(g) Observation 7

(h) Observation 8

(i) Observation 9

(j) Observation 10

Figure S3: Learned intensity functions for the spike patterns from monkey cortex.

**S.III.B.2  Human cortex, single neuron dynamics**

Table S4 provides the hyperparameters used for the dynamics recovery of the spike trains of the single, human cortex neuron. Note that the small value $\delta_0 = 1$ allows for the existence of both a self-exciting and a self-inhibiting behavior. The small value for $\lambda_\delta = 0.1$ indicates a fast decaying self-exciting kernel. On the other hand, the large value for $\lambda_\gamma = 100$ indicates a slowly decaying history kernel. The large value for $\mu_0 = 10$ and the small value for the excitation threshold $\alpha_0 = 2$ mediate the thinning effects.

Table S4: Hyperparameters for the Human Cortex, Single-Neuronal Experiment

| Function | Parameter | Value |
|---|---|---|
| $\lambda_n(t)$ | $\alpha_e$ | 1 |
| | $\beta_e$ | 1 |
| | $\alpha_m$ | 0.6 |
| | $\beta_m$ | 1 |
| | $\lambda_\delta$ | 0.1 |
| $h(t)$ | $\alpha_c$ | 10 |
| | $\beta_c$ | 1 |
| | $\lambda_\gamma$ | 100 |
| $\boldsymbol{w}_n$ | $\mu_0$ | 10 |
| | $\sigma_0$ | 10 |
| | $\nu_\tau$ | 1 |
| | $\alpha_\tau$ | 10 |
| | $\beta_\tau$ | 1 |
| | $\nu_\mu$ | 10 |
| | $\alpha_\mu$ | 0.001 |
| | $\lambda_\mu$ | 10 |
| $\phi_\mu(\alpha)$ | $\alpha_0$ | 2 |
| | $\delta_0$ | 1 |
| $\phi_\tau(\alpha)$ | $\alpha_0$ | 2 |
| | $\delta_0$ | 1 |

These patterns are further corroborated by Figure S4. Due to the fast decaying triggering intensity $\lambda^*(t)$, short-lived spikes are observed. Although the history kernel is slowly decaying, due to the scarcity of past spikes the aggregated history has limited effect on $\lambda(t)$. However, in some spike trains, thinning effects are observed.

(a) Observation 1

(b) Observation 2

(c) Observation 3

(d) Observation 4

(e) Observation 6

(f) Observation 6

(g) Observation 7

(h) Observation 8

(i) Observation 9

(j) Observation 10

Figure S4: Learned intensity functions for the spike patterns from human cortex.

Figure S5 illustrates the simulation of the learned MR-PP for the interval $[0, 80]$. The activity remains stable and similar to the physiological spike train that was used for the training.

(a) Simulated activity in [0,10]    (b) Simulated activity in [10,20]    (c) Simulated activity in [20,30]    (d) Simulated activity in [30,40]

(e) Simulated activity in [40,50]    (f) Simulated activity in [50,60]    (g) Simulated activity in [60,70]    (h) Simulated activity in [70,80]

Figure S5: Observed simulated activity of the MR-PP learned from human cortex spike patterns.

## S.III.C Details of the experimental results on the multi-neuronal spike train data

Table S5 provides the hyperparameters used for learning a MR-PP from the 25-neuron spike trains. The small value for $\lambda_\delta = 0.1$ indicates fast-decaying mutually triggering intensities and is justified by the presence of short spike bursts. The large value for $\delta_0 = 10000$ indicates a distinguishable relationship between a pair of types. Note that the threshold for the endogenous excitation $\alpha_0$ is different for $\phi_\mu(\alpha)(\alpha_0 = 0.001)$ and $\phi_\tau(\alpha)(\alpha_0 = 5)$. This means that for the endogenous intensity rates $\alpha_{m,n}$ in the range $[0.001, 5]$, the corresponding weight $w_{m,n}$ may be sampled from a Gaussian with zero mean but large variance. In turn, this fact can result in a positive $w_{m,n}$ indicating an additional excitation effect from type $m$ on type $n$ that may cancel repulsive effects from other types.

Table S5: Hyperparameters for the Multi-Neuronal Experiment

| Function | Parameter | Value |
|---|---|---|
| | $\alpha_e$ | 0.1 |
| | $\beta_e$ | 0.1 |
| $\lambda_n(t)$ | $\alpha_m$ | 10 |
| | $\beta_m$ | 1000 |
| | $\lambda_\delta$ | 0.1 |
| | $\alpha_c$ | 1 |
| $h_m(t)$ | $\beta_c$ | 100 |
| | $\lambda_\gamma$ | 5 |
| | $\mu_0$ | 10 |
| | $\sigma_0$ | 1 |
| | $\nu_\tau$ | 1000 |
| | $\alpha_\tau$ | 1 |
| $\boldsymbol{w}_n$ | $\beta_\tau$ | 10000 |
| | $\nu_\mu$ | 100 |
| | $\alpha_\mu$ | 0.001 |
| | $\lambda_\mu$ | 10 |
| $\phi_\mu(\alpha)$ | $\alpha_0$ | 0.001 |
| | $\delta_0$ | 10000 |
| $\phi_\tau(\alpha)$ | $\alpha_0$ | 5 |
| | $\delta_0$ | 10000 |

Our proposed algorithm is implemented in C++. All the tests are run on a Linux machine (Ubuntu 18.04 LTS) with CPU as Intel(R) Xeon(R) CPU E5-4627 v2, 3.30GHz, and 500GB memory.

Figure S6 illustrates the computational demands of the main steps of the inference algorithm. We plot the time for:

1. the Gibbs update of the parameters of the Hawkes intensity functions $\lambda_n^*(t)$ including the sampling of the events' parents.

2. the Gibbs updates of the parameters of the thinning probabilities $p_n(t)$, including the parameters of the history kernel $h_m(t)$.

3. the sampling of the thinned events.

The total inference time accounts also for the computation of the events' history and the sampling of their latent Pólya-Gamma variables, that are not plotted separately. This time reflects execution time among 5 threads for the steps of the inference that can be executed in parallel.

Figure S6: Inference Time of a MR-PP from the Multi-Neuronal Spike Trains.