[Reviews · NeurIPS 2019]

Reviewer 1



In the manuscript entitled, "Mutually Regressive Point Processes", the authors present a hierarchical model and associated Gibbs sampler for a modification of the mutually-excitatory Hawkes process to allow suppressive behaviour as well. The presentation is clear and both the proposed model and sampling scheme appear correct. The application of the model to a neural spike dataset is both topical (for the conference name! but really as a topic of current work) and interesting (since one must imagine the true model is not exactly of the class assumed: i.e., application with [presumably] a typical degree of misspecification). I think this model has the potential to be applied to a number of scientific problems in addition to the neural model: for instance, in studies of disease where new cases spread and generate others, but each can also increase the likelihood of a community receiving prophylaxis in response. Given the level of detail in the current manuscript, the examination of both well-specified and misspecified (real world) data, and the supplementary material of the code, I would consider this paper already completely sufficient to warrant acceptance. That said, I would encourage the authors to think about some issues for future work / if planning to release the code through (e.g.) an R package. One is obviously scalability and approximate posterior representation as the authors themselves allude to. The other is prior choice. These deep hierarchical models can often be difficult to choose good priors for on a case by case basis, yet very broad 'non-informative' priors can nevertheless pull the model away from the data-preferred regions of parameter space in surprising/unwanted ways. Ideally the authors could come up with a set of summary statistics or visualisations that would be appropriate to help new users choose appropriate hyper-parameters for their priors and/or diagnose prior-sensitivity. (The latter might be indicable from posterior re-weighting; if Radon-Nikodym derivatives for the collapsed processes can be calculated a la Ghosh et al. 2006 for the Dirichlet Process).

Reviewer 2



Main comments 1) When the model is introduced in section 2.3 I would have liked to have read some discussion about why this structure would be beneficial and sufficiently flexible to do the job that you're asking of it. I figured it out slowly, but it took probably longer than it would have with a bit of author assistance. 2) Relatedly, after the model (and prior) have been introduced fully, it would have been nice to see some kind of demonstration of the range and kinds of excitation/inhibition behaviour that is now possible to both simulate and model. What is this model capable of doing? Where are its (simulatory) limitations etc? Show how it goes beyond the most relevant competing models. 3) There were no real comparisons to other methods. So I don't know if I am getting anything extra in the model, compared to other approaches out there, and if the only benefit here is the exact simulation component. If there is nothing extra in the model compared to other approaches, this is fine, but it should be made clear, and that it is the computational contribution that is primary here. On the other hand, if there is greater flexibility in the new model than existing approaches, then the authors should not hold back from actually demonstrating this! 4) Figure 4(b) is clearly bimodal, with both positive and negative modes. There is also evidence that many of the other parameters are multi modal. What does this mean in terms of understanding the excitation/inhibition process? Alternatively, as this is inference based on data simulated under the model, is this just evidence that the Markov chain sampler has not adequately explored the posterior yet, in which case why don't you run the chain for longer and provide a better description of the (marginal) posteriors? More generally, given that exact computation for this continuous time model is important, I would have liked to have seem more discussion on the sampler performance in terms of mixing (e.g. integrated autocorrelation time, autocorrelations, or whatever is suitable). Currently given the bumps in Figures 1 and 2 I'm slightly suspicious of sampler performance. Exact simulation is good, but if performance is terrible compared to approximate methods, then ... 5) L.207 The simulated activity in Fig 4(c) is stated to "remain psychological and similar to the one used for training fin Fig 4a." This is making a statement about the application-specific based structure of the data that is not properly quantified or qualified. What makes this particular simulated data "better" or more relevant in an application-based sense than those that could have come from other methods? This needs explaining/quantifying more clearly. Minor Comments The paper would benefit from a close reading of sentences for grammar and structure. Please use comma's where it would be beneficial to understand the phrasing of a sentence, for example. L.142, what does the asterisk denote in the numerator of the LHS fraction? It was mildly annoying that much of the information that could have been in the main paper (with a little consideration of space restrictions) was instead relegated to supporting information, requiring forward-backward flipping constantly. Maybe the authors could consider if any important details could be returne to the main paper to minimise this to the less important parts?

Reviewer 3



A generally good paper. The paper proposes a class of point processes, called mutually regressive point processes that captures both excitation and inhibition in sequences of events. The proposed approach, can model nonlinear temporal interactions in continuous time. The construction builds on a generalization of Hawkes Processes in a new combination of existing methods. We can recover the traditional Hawkes processes by certain parameter settings. The resulting construction is quite involved. It is based on the augmentation of the Hawkes Process intensity with a probability term. This term is equipped with a weight parameter that tunes the influence of a certain type of event to another. In particular, a sparse normal gamma prior is proposed to be placed on the weights so that an interaction between 2 types is inhibitory or excitatory in a probabilistic manner. The paper is well justified and supported for the sampling scheme in the generative model as well as the inference procedure. For the former, there is a detailed explanation for the hierarchy, the parameters and the constraints. For the latter, there is a thorough description of the inference and the related methods (eg thinning) However, as a downside the paper is heavy in notation which is because of the many parameters involved in the construction. The heavy notation makes the reading/comprehension of the paper not straightforward. The authors give the precise algorithms for inference. The synthetic data experiment supports the validity of the model. Regarding other experiments, although the paper brings interesting directions in the use of these processes for neuron data, the experiments are limited. Also, sufficient comparisons to other models on both interpretation and predictive performance are missing.

[Author Response · NeurIPS 2019]



| (a) Revised Figure 1.b | (b) Revised Figure S1.b | (c) Revised Figure 6.b | (d) MAUP for Sec 4.3 |

Figure 1: *Revised experiment results for Mutually Regressive Point Processes.*

We would like to thank the reviewers for their detailed and constructive reviews of our manuscript. Their comments
definitely help us clarify many points in the paper. We did our best to address the main areas of concern: i) convergence
of the learning algorithm. We ran the learning algorithm for more MCMC iterations, and we revised the relevant Figures
1a, 1b); and ii) limited experiments. We compare against PP-GLMS (discrete-time model capable of capturing general
temporal interactions) and Hawkes processes (continuous time, capable of capturing only excitatory interactions)
(Figure 1c). We also demonstrate experimentally the sensitivity of the PP-GLM parameters to the boundaries used for
the temporal aggregation resulting in a different sign of effect (from excitatory to inhibitory) in Figure 1d.

**Convergence of the learning algorithm (Reviewer 2 - comment 4).** We reran the learning algorithm for 5000, instead
of 1000 MCMC iterations for the synthetic experiment. In Figure 1a, we plot the new posterior obtained. In Figure
1b, we plot the autocorrelation (initially provided in the supplementary material) for that parameter. As was correctly
pointed out, the two modes were merged into one at 0 after running the MCMC for more iterations. The oscillations of
the autocorrelation around 0 were also reduced after a larger number of samples. The modes in the initial manuscript
did not raise a warning flag, since they were insignificant: both of them were close to zero (compared to the very large
negative weights for the rest of interactions), indicating no considerable effect from type I coming from the non-linear
part of the intensity, as dictated by the prior and due to the self-excitation. The small bumps disappeared for the rest of
the parameters. (We will update the camera-ready accordingly in case of acceptance). **Additional experimental results**
**(Reviewer 2 - comment 3, Reviewer 3).** In Figure 1c, we provide the test-logl not only of the PP-GLMs (typically used
for spiking data), but also that of a Hawkes process (HP), indicating that loss in the generalization capability of the model
stems mostly from the time discretization, although capturing inhibitory effects could further improve its performance.
We illustrate one weakness of the discrete-time PP-GLM models briefly discussed in the introduction ("However, the
estimated regression coefficients may vary widely depending on the boundaries chosen for aggregation"), known as the
Modifiable Areal Unit Problem (MAUP) [1] that is attributed to the temporal aggregation of the spikes. In Figure 1d,
we plot the coefficients of the PP-GLM learned for variable size of the time-bin (assuming unit degree of regression).
For the effect from neuron 23 on neuron 1, for example, the sign of the interaction changes from positive (for 0.1
msec and 0.3 msec) to negative (for 0.2 msec), indicating a potentially time-varying (in terms of its sign), relationship.
Although, this problem could potentially be solved by considering a degree of regression larger than one and finer
time-bins, these parameters have to be predetermined (potentially for each neuron/ type separately). Moreover, the
change-points may depend dynamically on the temporal history. For that particular experiment, different configuration
of time-bin size and degree of regression did not yield improvement in terms of the predictive likelihood. The proposed
continuous-time model inherently circumvents these limitations by allowing i) two channels of time-varying interaction
one coming from the mutually exciting intensity function, the other from the sigmoidal part which may or may not be
mutually exclusive by adjusting the parameters of the prior accordingly ii) the parameters that regulate the excitatory
and inhibitory effects to be learned dynamically from the data. We will include a network illustrating the temporal
interactions identified by the proposed model and the MAUP for the rest of the neurons in the camera-ready in case
of acceptance. **Importance of stable dynamics (Reviewer 2 - comment 5).** One immediate advantage of a model
with stable dynamics, is that it allows long-run time predictions. We refer to the paper [2] (Section "Importance of
stable point process models for applications"), for a detailed explanation on the importance of obtaining physiological
temporal patterns. **Typo L142 - (Reviewer 2).** Thank you for pointing out the typo in L142: the LHS is identical to the
product term in the RHS of Equation (14), L138. **Prior Choice - (Reviewer 1).** We thank Reviewer 1 for the positive
comments.We will incorporate your feedback (along with suggestions for the prior choice, e.g empirical Bayes) in the
discussion section of the camera-ready version in case of acceptance. Based on our current experiments, especially the
parameters in the activation functions are critical and hence worth being finely tuned.

**[1]** Fotheringham, A. S., Wong, D. W. (1991). The modifiable areal unit problem in multivariate statistical analysis.
Environment and planning A, 23(7), 1025-1044. **[2]** Gerhard, F., Deger, M., Truccolo, W. (2017). On the stability
and dynamics of stochastic spiking neuron models: Nonlinear Hawkes process and point process GLMs. PLoS
computational biology, 13(2), e1005390.


[Meta-Review · NeurIPS 2019]

This article proposes a novel continuous-time model for interacting time-event data. The model is able to capture both excitatory and inhibitory interactions. A complete Gibbs sampler is described for posterior inference. The paper is well written. The experimental results are interesting, although the predictive performances do not demonstrate a huge gain in using the proposed model compared to simpler alternatives (even considering the additional experiments provided by the authors in the response).